# Haloperidol bound D$_2$ dopamine receptor structure inspired the discovery of subtype selective ligands

Luyu Fan[1,4], Liang Tan[2,4], Zhangcheng Chen[1], Jianzhong Qi[1], Fen Nie[1], Zhipu Luo[3], Jianjun Cheng [2✉] & Sheng Wang [1✉]

The D$_2$ dopamine receptor (DRD2) is one of the most well-established therapeutic targets for neuropsychiatric and endocrine disorders. Most clinically approved and investigational drugs that target this receptor are known to be subfamily-selective for all three D$_2$-like receptors, rather than subtype-selective for only DRD2. Here, we report the crystal structure of DRD2 bound to the most commonly used antipsychotic drug, haloperidol. The structures suggest an extended binding pocket for DRD2 that distinguishes it from other D$_2$-like subtypes. A detailed analysis of the structures illuminates key structural determinants essential for DRD2 activation and subtype selectivity. A structure-based and mechanism-driven screening combined with a lead optimization approach yield DRD2 highly selective agonists, which could be used as chemical probes for studying the physiological and pathological functions of DRD2 as well as promising therapeutic leads devoid of promiscuity.

[1] State Key Laboratory of Molecular Biology, CAS Center for Excellence in Molecular Cell Science, Shanghai Institute of Biochemistry and Cell Biology, Chinese Academy of Sciences, University of Chinese Academy of Sciences, 320 Yueyang Road, 200031 Shanghai, China. [2] iHuman Institute, ShanghaiTech University, 393 Middle Huaxia Road, 201210 Shanghai, China. [3] Institute of Molecular Enzymology, Soochow University, 215123 Suzhou, Jiangsu, China. [4] These authors contributed equally: Luyu Fan, Liang Tan. ✉email: chengjj@shanghaitech.edu.cn; wangsheng@sibcb.ac.cn

G-protein-coupled receptors (GPCRs)—the most intensely investigated drug targets in the pharmaceutical industry— regulate numerous diverse physiological processes and have druggable sites that are accessible at the cell surface[1]. Correspondingly, ~34% of US Food and Drug Administration (FDA)-approved drugs act primarily through them[2]. Unfortunately, many of the GPCR ligands that are used as drugs or pharmacological tools are not selective and exhibit some unintended activity on nontarget GPCRs or other proteins[3]. Dopamine receptors belong to the GPCR superfamily and are divided into two subfamilies on the basis of sequence similarity and pharmacological profiles. The $D_1$-like receptors (DRD1 and DRD5) promote intracellular cAMP accumulation through activating $G\alpha_s$ or $G\alpha_{olf}$ proteins[4]. In contrast, $D_2$-like receptors (DRD2, DRD3, and DRD4) activate $G\alpha_{i/o}$ proteins to diminish cAMP levels as well as modulate certain ion channels[4]. DRD2 is arguably one of the most well-established drug targets in neurology and psychiatry. For instance, most receptor-based antiparkinsonian drugs work via stimulating the DRD2, whereas all FDA approved antipsychotics are well-known DRD2 antagonists or partial agonists[5]. Medications that target DRD2 are also used to treat hyperprolactinemia, restless legs syndrome, Tourette's syndrome, among many other disorders. So far, however, there is no truly selective DRD2 ligands[6–10]. Most DRD2 ligands concomitantly bind to the DRD3 or/and DRD4[6–10]. Thus, there is a desire to develop compounds that selectively target the DRD2 with minimal subtype cross-reactivity, and to ascertain the physiological and pathological functions governed by DRD2.

The discovery of selective DRD2 ligands has been challenging[6]. This is not surprising given that the sequence similarities of the transmembrane (TM) regions are 53% for DRD2 versus DRD4 and 78% for DRD2 versus DRD3[4]. As a result, the orthosteric-binding pockets (OBPs), where the majority of dopaminergic ligands bind are quite similar among $D_2$-like receptor subtypes. Although substantial efforts have led to the discovery of DRD3-selective and DRD4-selective ligands[9], significantly less progress has been made toward highly DRD2-selective compounds[6–8,10]. Recently, discovery campaigns have been catalyzed by structure-based drug design (SBDD)[11]. Owing to the identification of the unique rigid extended binding pocket (EBP) for each receptor, several DRD3-selective and DRD4-selective ligands have been identified via SBDD in the last few years[12–14].

We previously solved the structure of DRD2 in complex with the atypical antipsychotic drug-risperidone and identified the EBP of DRD2[15]. However, this EBP is not a rigid pocket as those of DRD3 and DRD4[13–16]. The success rate of SBDD is much lower if the target binding pocket is not rigid[17], just like the DRD2 EBP. To address this problem, we solve here the complex structure of the DRD2 bound to a commonly used typical antipsychotic drug-haloperidol. Haloperidol is a potent antagonist of the DRD2 and it shares a substructure with the reported DRD2-preferring compound L-741626[6] (Fig. 1a). Analysis of the DRD2–haloperidol complex structure reveals an unexpected second extended binding pocket (SEBP). Significantly, we find that the SEBP not only directly interacts with the haloperidol, but also plays a key role in DRD2 agonist activation. Driven by our structural delineation of the unique ligand-binding pose at DRD2 and activation mechanism via SEBP and OBP, we further obtain two DRD2 subtype-selective agonists—**$O_4SE_6$** and **$O_8LE_6$**, excluding agonism at DRD3 and DRD4.

## Results

**Insights from the DRD2/haloperidol structure**. To the best of our knowledge, there are only a few DRD2-preferring compounds reported to date[6]. These include the Merck compound L-741626,

which shows around 10-fold DRD2 versus DRD3/DRD4 selectivity in radioligand-binding assays (Fig. 1a). To obtain structural insights into the unique ligand-binding pocket of DRD2, the same T4 lysozyme (T4L) insertion construct as the one previously engineered to obtain DRD2/risperidone complex structure was used[15]. The L-741626 were then screened in crystallization trials. Although we were able to obtain small complex crystals, the quality of these crystals could not be further improved through the use of additives and other condition optimizations. Then, the commonly used typical antipsychotic drug-haloperidol (Fig. 1a), which shares a similar chemical structure with L-741626, was screened in crystallization trials. We eventually obtained the crystal structure of the DRD2/haloperidol complex at a resolution of 3.1 Å (Fig. 1b, Supplementary Fig. 1a–e and Supplementary Table 1). Haloperidol is anchored to DRD2 by a conserved salt bridge between the protonated nitrogen in the piperidine ring and the conserved aspartate, $Asp114^{3.32}$ (superscripts represent the Ballesteros–Weinstein residue numbering[18]) —a canonical interaction for aminergic and many other GPCRs[13,16,19] (Supplementary Fig. 1f).

Comparison of the DRD2/haloperidol and DRD2/risperidone crystal structures reveals an overall 1.5–2 Å binding pocket compaction with an outward shift of the extracellular tip of TM1 and an inward shift of the extracellular tip of TM2 (Supplementary Fig. 1g). The volume of EBP in the haloperidol-bound structure is significantly reduced when compared to that of the risperidone-bound structure (Fig. 2a, b). This is likely due to the more compact positions of TM2 and TM7 around the ligand in the haloperidol-bound DRD2 structure, and the inward rotation of EBP key residues: $Glu95^{2.65}$ and $Tyr408^{7.35}$ (Supplementary Fig. 1g, h). The chlorobenzene moiety of haloperidol reaches closer to the cleft between TM2 and TM3 (Fig. 1c–f) and extends much further toward extracellular loop (EL)1, whereas the terminal of risperidone makes an aromatic interaction with the top turns of $TM7^{15}$ (Fig. 1c–f). Notably, the conserved residue $Trp100^{EL1}$ in DRD2/haloperidol structure rotates outward away from the binding pocket as compared to the risperidone-bound structure (Fig. 1d–f). Similar crystal contacts between the extracellular tip of TM3 and the symmetry-related T4L at the DRD2/haloperidol and DRD2/risperidone crystal structures were observed, but there is no crystal contact between $Trp100^{EL1}$ and the symmetry-related T4L at both structures (Supplementary Fig. 2a, b). Therefore, the rotation of $Trp100^{EL1}$ at DRD2 is unlikely induced by crystal packing forces. Although the electron-density omit map partially missed at the chlorobenzene moiety of the haloperidol (Supplementary Fig. 1c, d), haloperidol apparently prevents the inward rotation of $Trp100^{EL1}$ (Fig. 1d, e), which may explain the difference between the two structures. And, the mutations of $Trp100^{EL1}$ to Phe or Ala in DRD2 decreased the binding affinity of haloperidol and L-741626 (Supplementary Table 2).

**The distinct SEBP of DRD2**. The outward rotation of $Trp100^{EL1}$ in the haloperidol complex allows the formation of a SEBP, which is occupied by the chlorobenzene moiety of haloperidol (Figs. 1b–e and 2a). This rearranged DRD2 SEBP consists of residues from TM2, TM3, EL1, and EL2 and is defined by $Trp100^{EL1}$ and $Phe110^{3.28}$ (Figs. 1b, 2a and Supplementary Fig. 1e). The DRD2 SEBP in risperidone-bound structure is disrupted, due to the inward rotation of $Trp100^{EL1}$ (Figs. 1d, f and 2b). Although the conserved residue $Trp^{EL1}$ of DRD3 and DRD4 locates in the same position as that in DRD2/haloperidol complex structure, the inward movement of EL2 in DRD3 and DRD4 forms a border of EBP in each receptor (Fig. 2c, d and Supplementary Fig. 3a–e). And, the different position of EL2 at $D_2$-like

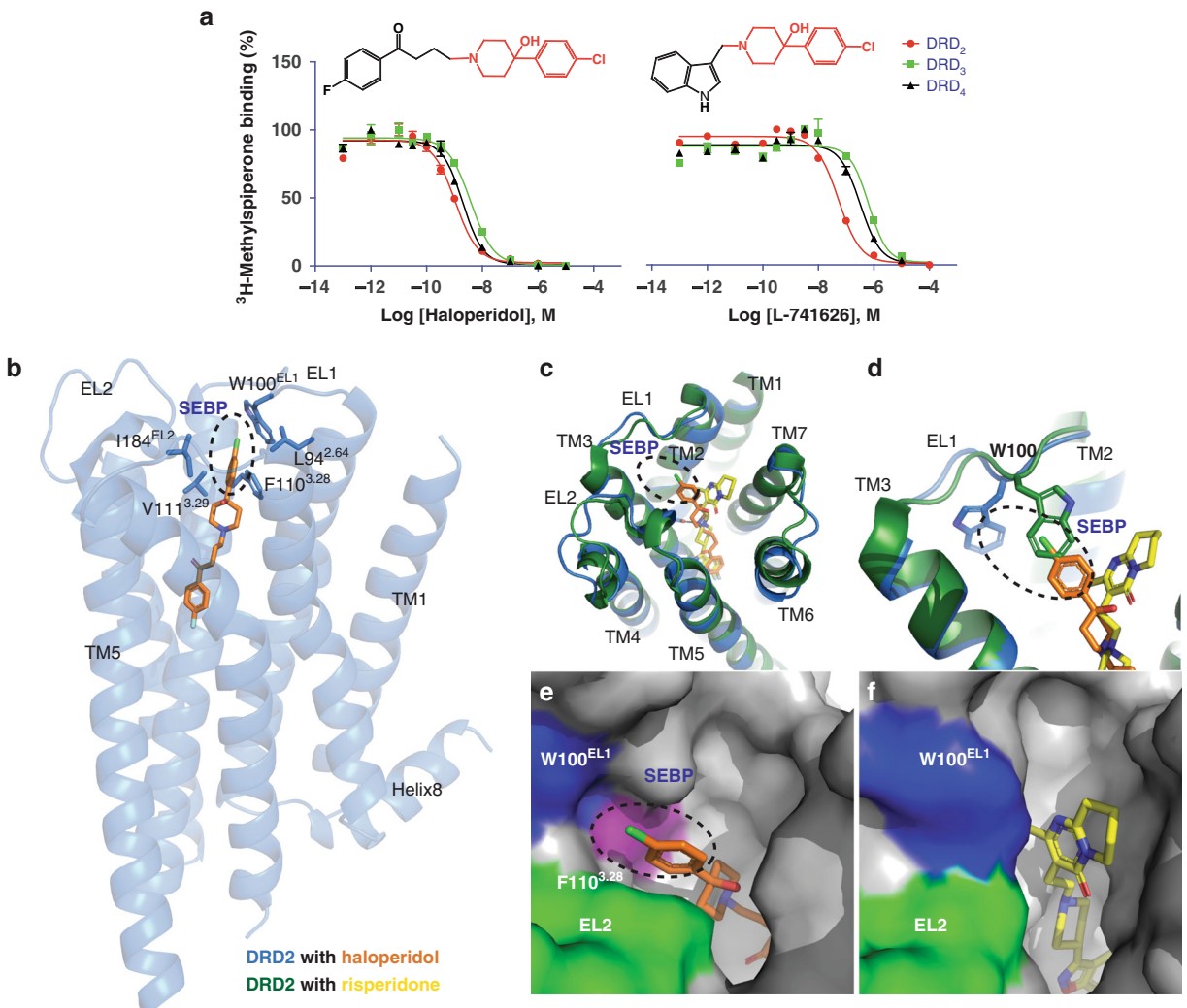

**Fig. 1 The architecture of second extended binding pocket at DRD2. a** Binding affinity of haloperidol and L-741626 in $D_2$-like receptor subtypes. Error bars, SEM ($n = 3$ independent experiments). See also Supplementary Table 2. Source data are provided as a Source Data file. **b** Overall structure of the DRD2/haloperidol complex. The receptor is shown in blue cartoon, haloperidol in orange sticks, and the residues of SEBP (ellipse) in blue sticks. **c**, **d** Comparison of the view from the extracellular side of a structural alignment DRD2/haloperidol (blue cartoon/orange stick) and DRD2/risperidone complex (green cartoon/yellow stick; PDB code: 6CM4). The salt bridge interaction between DRD2 and haloperidol is shown as gray dashed line. **e**, **f** Top views of the extended binding pocket in the DRD2/haloperidol **e** and DRD2/risperidone **f** complexes. The receptor pocket surface is colored gray except the EL2 in green, W100[EL1] in blue and F110[3.28] in purple. Ligands are shown as capped sticks with carbons colored orange (haloperidol) and yellow (risperidone). In all panels, the Ballesteros–Weinstein numbering is shown as superscript. The position of SEBP is shown as an ellipse.

receptors is unlikely induced by the crystal packing forces (Supplementary Fig. 2c, d). The DRD2 SEBP partially overlaps with the previous identified DRD3 EBP, which consists of TM2, TM7, EL1, and EL2[16] (Fig. 2a, c). Compared to the DRD3, the outward movement of EL2 makes additional space for the SEBP at DRD2 (Fig. 2c, d and Supplementary Fig. 3c).

To further identify the key residue(s) responsible for the binding of DRD2-preferring compounds, we performed mutagenesis studies on the SEBP-related residues (Supplementary Table 2 and Supplementary Fig. 3f). The alanine substitution of most SEBP residues, except Phe110[3.28], slightly reduced the affinity of both haloperidol and L-741626 (Supplementary Table 2). The mutation of Phe110[3.28] to Ala or Leu on DRD2 greatly enhanced the binding of haloperidol or L-741626 (15.33 or 1.77-fold for haloperidol and 144.18 or 18.65-fold for L-741626) (Supplementary Table 2), while the mutation of Phe110[3.28] to Trp or Tyr greatly reduced the binding of haloperidol and L-741626 (28.00 or 41.41-fold for haloperidol

and 8.51 or 57.54-fold for L-741626) (Supplementary Table 2). Furthermore, the mutation of Phe110[3.28] to Cys or Glu, both of which have similar sizes with each other but with different physical properties, slightly influenced the binding affinity of both ligands, ruling out the possibility that the property of the amino acid affects ligand binding (Supplementary Table 2). It is possible that the alanine or leucine substitution of Phe110[3.28] makes additional space for the DRD2 SEBP, facilitating the accommodation of the chlorobenzene moiety of haloperidol or L-741626 (Figs. 1b, e and 2a, e). And, the previous published studies already showed that the mutation of Phe110[3.28] to Ala on DRD2 did not enhance the binding of non-selective compounds—risperidone and nemonapride[15]. In summary, the size of the residue 3.28 seems to play a key role for the binding affinity of haloperidol and L-741626.

Different from DRD2 and DRD3, the bulky residue Phe[3.28] is replaced by leucine in the homologous position of DRD4, which makes extra space for the DRD4 EBP[13] (Fig. 2e–h). Through

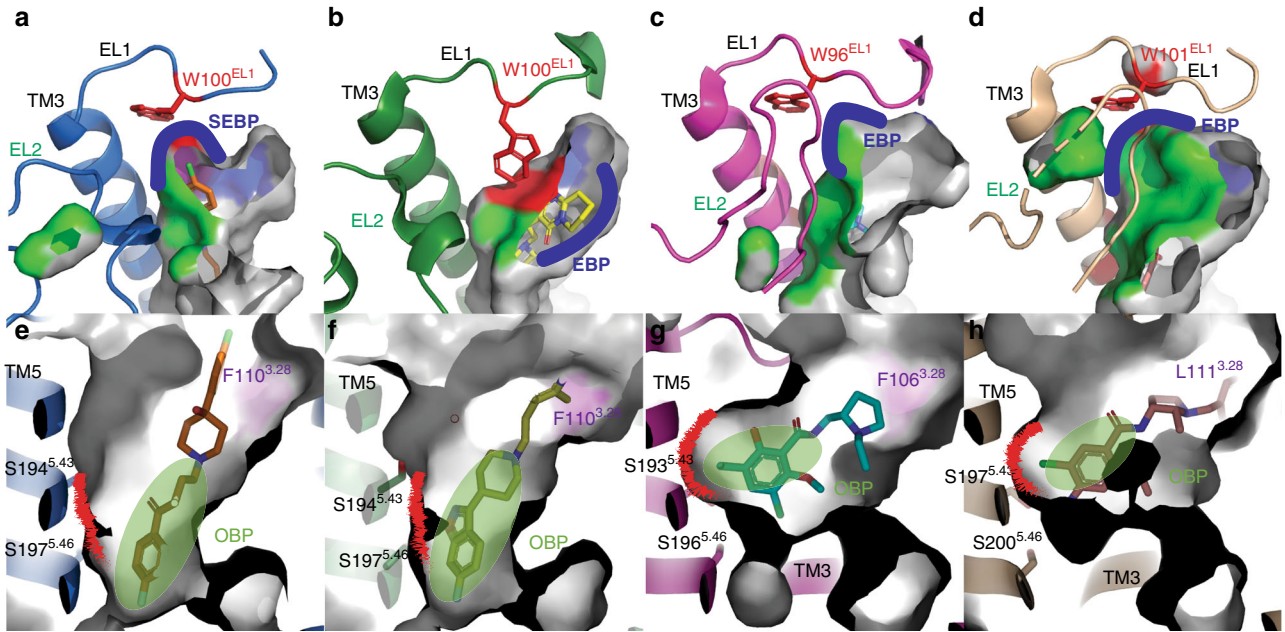

**Fig. 2 Comparison of the binding pocket across the D2-like family receptors. a, e** DRD2/haloperidol (blue cartoon/orange stick). **b, f** DRD2/Risperidone (green cartoon/yellow stick, PDB code 6CM4). **c, g** DRD3/eticlopride (magenta cartoon/cyan stick, PDB code: 3PBL). **d, h** DRD4/nemonapride (tan cartoon/pink stick, PDB code: 5WIU). In all panels, receptors are shown as surface or cartoon. Ligands are shown as sticks. Surface representations of the ligand-binding pocket. The surface of residue 3.28 is colored as purple, EL2 is colored as green and W[EL1] is colored as red. The border of SEBP or EBP is marked as blue and OBP is marked as red. Ballesteros–Weinstein numbering is shown as superscript.

mutating the less bulky Leu111[3.28] to bulky phenylalanine thus shrinking the volume of DRD4 EBP, the binding affinities of haloperidol and L-741626 to DRD4 decreased (2.04-fold for L-741626 and 10.71-fold for haloperidol) (Supplementary Table 2). Further, the binding affinities of haloperidol and L-741626 to Phe106[3.28]Ala mutant of DRD3 increased (34.67-fold for L-741626 and 1.99-fold for haloperidol) (Supplementary Table 2). In contrast, haloperidol and L-741626 were shown to have greater affinity enhancement at Phe110[3.28]Ala mutant of DRD2 (144.18-fold for L-741626 and 15.33-fold for haloperidol) than DRD3 compared to wildtype receptors, respectively (Supplementary Table 2). These results suggest that the residue Phe[3.28] may play a key role in DRD2 versus DRD3/DRD4 selectivity.

**Structure inspired discovery of selective DRD2 ligands.** The structural determination of the DRD4 EBP, defined by Phe91[2.61] and Leu111[3.28], enabled the structure-based discovery of compounds that are highly specific for this receptor[13,14]. In the recently published paper[14], we docked over 138 million molecules against the EBP and OBP of DRD4. In our selected 549 make-on-demand molecules, which covered high-ranking (−75 to −63 kcal mol−1), mid-ranking (−61 to −46 kcal mol−1) and low-ranking compounds (−43 to −35 kcal mol−1), 81 compounds (54 compounds from high-ranking scores; 27 compounds from middle-ranking scores) were shown to have DRD4 affinity and 468 compounds (164 compounds from high-ranking scores; 164 compounds from middle-ranking scores; 140 compounds from low-ranking scores) failed to bind to DRD4 (Supplementary Fig. 4)[14]. In these 81 DRD4-binding compounds, six compounds showed binding affinities for all three D2-like receptors[14], and two compounds displayed binding affinities for DRD4 and DRD3[14]. Although the SEBP or EBPs of D2-like receptors present critical differences, the OBPs of D2-like receptors locate in a similar position of each receptor and partially overlap with each other (Fig. 2e–h). The overall similarity of ligand-binding pockets may

explain the facts that the eight high/mid-ranking DRD4-bound compounds could concomitantly bind to DRD3 or/and DRD2 as well[14], and those 140 low-ranking DRD4 compounds did not bind to DRD3 and DRD2 either (Supplementary Fig. 4).

Although the locations of the OBPs of D2-like receptors are very similar, their shapes are strikingly different between DRD2 and DRD3/DRD4 (Fig. 2e–h). Compared to the DRD3 and DRD4, the inward shift of TM5 in DRD2 shrinks its OBP substantially (Supplementary Fig. 5a–d and Fig. 2e–h). As a result, although the ligands bind to the same pocket-OBP, their orientations are completely different, with only partial overlap, between DRD2 and DRD3/DRD4 (Fig. 2e–h). The ligand in DRD2 is located deeper in the OBP and embeds in the deep binding pocket defined by the side chains of TM3, TM5, and TM6, which accommodates the butyrophenone moiety of haloperidol (Fig. 2e) and benzisoxazole moiety of risperidone (Fig. 2f)[15]. By contrast, the ligands in DRD3 and DRD4 are located higher in the OBP, pointing to TM5, adopting a shallow binding mode (Fig. 2g, h). In the recent published paper, the L-745870 which shares a similar chemical structure with haloperidol and L-741626, also adopts a similar shallow binding mode, not the deep binding mode, in the DRD4/L-745870 complex structure[20]. Molecular docking of eticlopride and nemonapride to the structure of DRD2 showed that these ligands also adopt a similar binding pose as haloperidol or risperidone in the complexes (Supplementary Fig. 5e, f). Even though the molecular sizes of eticlopride and nemonapride are obviously smaller than those of haloperidol and risperidone, they failed to bind to the receptor in an orientation analogous to those poses in the DRD3 and DRD4 structures, respectively. This is likely a direct consequence of the main movement of TM5, which consequently affects the size and shape of the OBP, allowing eticlopride and nemonapride to engage a deep binding pose at DRD2 (Supplementary Fig. 5e, f).

Based on both the similarity of the OBPs of DRD2, DRD3, and DRD4, as well as the distinct EBPs of these receptors, we anticipated

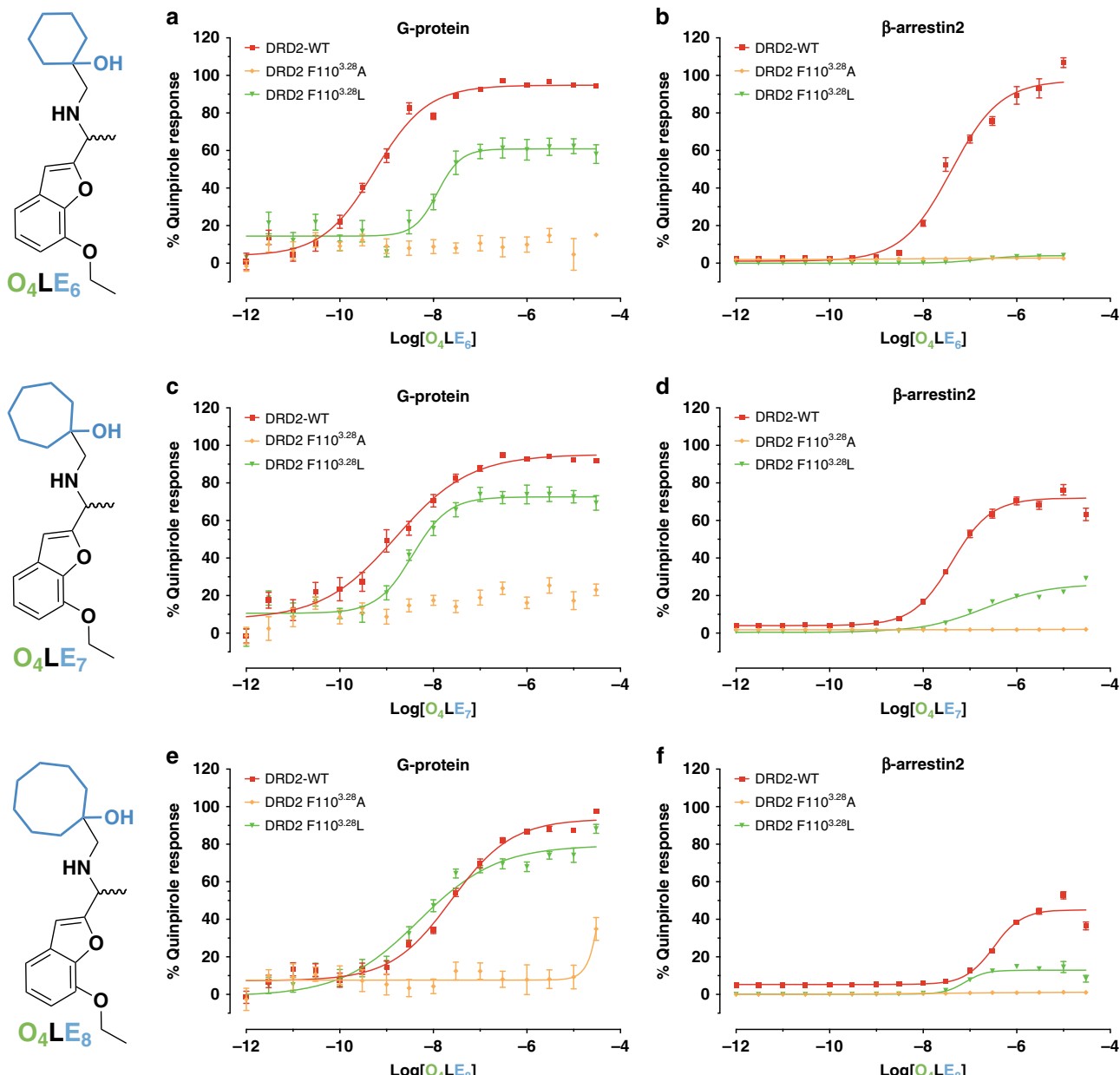

**Fig. 3 Mutation of Phe110³·²⁸ decreases the O₄LE₆₋₈'s efficacy at DRD2.** Profiling of **O₄LE₆₋₈** measuring DRD2 G protein activity ($G_{\alpha i/o}$-mediated cAMP inhibition; **a**, **c**, **e**) and β-arrestin2 recruitment (Tango; **b**, **d**, **f**), normalized to percent quinpirole activity. Data represent three independent experiments performed in triplicate technical replicates and in parallel using the same drug dilutions. Error bars, SEM ($n = 3$ independent experiments). See also Supplementary Tables 5 and 6. Source data are provided as a Source Data file.

that DRD2-preferring and/or DRD3-preferring compounds could potentially be identified from the abovementioned 164 high-ranking and 164 mid-ranking DRD4 unbound molecules. Indeed, four compounds were identified from these 328 compounds, which showed DRD2 and/or DRD3-binding affinities ($K_i$) ranging from 0.21 to 4.27 μM in a competition binding assay with the antagonist radioligand [³H]-N-methylspiperone (Supplementary Table 3 and Supplementary Fig. 4). Notably, compound **540595123** (**O₄LE₆**) showed binding affinity for DRD2 and DRD3 ($K_i = 1.91$ μM for DRD2/0.22 μM for DRD3, Supplementary Table 3), and also displayed potent agonist activity in a following DRD2 $G_{\alpha i/o}$-mediated cAMP inhibition assay (EC₅₀ = 0.57 nM, Fig. 3a and Supplementary Table 3).

Since the EC₅₀ value in the $G_{\alpha i/o}$-mediated cAMP inhibition assay may be influenced by the signal amplification, **O₄LE₆** was subjected to an orthologous nonamplification assay of DRD2 G protein activity measuring $G_{\alpha i}1$-γ2 dissociation by bioluminescent resonance energy transfer (BRET). In this assay, **O₄LE₆** showed modest agonist activity (EC₅₀ = 24.14 nM, Supplementary Table 3), recapitulating our findings obtained from measuring $G_{\alpha i/o}$-mediated cAMP inhibition activity. The use of the antagonist radioligand [³H]-N-methylspiperone could explain the difference between binding $K_i$ and functional EC₅₀ values, which is consistent with previous results demonstrating that the affinity of agonists for an uncoupled GPCR (traditionally been referred to as the 'low-affinity' state) would appear very low[10,21–23]. This 'low-affinity' state was also observed with the control compound dopamine in these assays, which showed a $K_i$ of 1.58 μM but an EC₅₀ of 0.20 nM ($G_{\alpha i/o}$-mediated cAMP inhibition assay) or 50.37 nM (BRET), respectively (Supplementary Table 3).

**The DRD2 activation mechanism via the SEBP**. As mentioned above, we confirmed that the residue Phe110$^{3.28}$ makes direct contact with haloperidol (Fig. 1b, e) and plays a key role in the DRD2-preferring antagonist binding, such as L-741626 (Supplementary Table 2). Further, we performed ligand-binding assays to characterize the pharmacological properties of the wild-type and Phe$^{3.28}$Ala-mutant receptors using DRD2 agonists–**OLE** compounds (Supplementary Note 1). The affinity of the Phe110$^{3.28}$Ala DRD2 mutant for **O₄LE₆** increased by 2.64-fold, with $K_i$ values of 1.90 and 0.72 μM for the wild-type and Phe110$^{3.28}$Ala-mutant receptors, respectively (Supplementary Table 3). In the case of Phe106$^{3.28}$Ala DRD3 mutant, different from previous haloperidol and L-741626, the affinity for **O₄LE₆** was decreased by 2.09-fold, with $K_i$ values of 0.22 and 0.46 μM for the wild-type and Phe106$^{3.28}$Ala-mutant receptors, respectively (Supplementary Table 3). Similar results were observed in **O₄LE₅**, **O₄LE₇**, and **O₄LE₈** (the affinity increased 10.27-fold, 7.52-fold, 8.47-fold for DRD2; decreased 4.19-fold, 1.75-fold, 1.62-fold for DRD3) (Supplementary Table 3). Taken together, these results indicate that the residue Phe$^{3.28}$ could be as a key indicator to distinguish DRD2 versus DRD3 selectivity.

To further test the hypothesis that the contact between **OLE** compounds and Phe110$^{3.28}$ in the SEBP facilitates DRD2 activation, we examined whether the Phe110$^{3.28}$ mutants were critical for Gα$_{i/o}$ signaling and β-arrestin2 recruitment activity for **O₄LE₆₋₈** (at Phe110$^{3.28}$Ala and Phe110$^{3.28}$Leu, respectively). In both mutants, DRD2 expression levels were comparable to that of the wild type (Supplementary Fig. 6c). With the Phe110$^{3.28}$Ala mutant, **O₄LE₆₋₈** failed to activate Gα$_{i/o}$ or to recruit β-arrestin2 (Fig. 3 and Supplementary Tables 5, 6), indicating that the Phe110$^{3.28}$Ala substitution disrupts both Gα$_{i/o}$ and β-arrestin2 agonism. Whereas the Phe110$^{3.28}$Leu substitution led to **O₄LE₆₋₈**'s partial activation of Gα$_{i/o}$ and β-arrestin2-signaling pathways with respect to full agonist quinpirole (Fig. 3, Supplementary Fig. 6a, b and Supplementary Tables 5, 6). In the Phe110$^{3.28}$Ala/Leu mutants, quinpirole showed similar agonist activity in both Gα$_{i/o}$ and β-arrestin2-signaling pathways as that in the wild type although with slightly reduced potency (EC$_{50}$ = 0.20, 3.39, and 4.04 nM for wild type, Phe110$^{3.28}$Ala and Phe110$^{3.28}$Leu in Gα$_{i/o}$ signaling, and EC$_{50}$ = 7.34, 210.20, and 44.17 nM in β-arrestin2 recruitment, respectively) (Supplementary Tables 5 and 6). When the cyclohexane substituent of **O₄LE₆** (which interacts with residue Phe110$^{3.28}$) was enlarged, such as in compounds **O₄LE₇** and **O₄LE₈**, agonist efficacy was partially rescued at the Phe110$^{3.28}$Leu DRD2 mutant compared to **O₄LE₆** (Fig. 3 and Supplementary Tables 5, 6). These results suggest that the recognition of the RHS of **O₄LE₆₋₈** in the DRD2 SEBP, specifically by Phe110$^{3.28}$, leads to an auxiliary mechanism of agonist activation via the SEBP.

**O₄LE₆** is a racemic mixture of **O₄SE₆** and **O₄RE₆** and the individual enantiomers were docked in order to understand their mechanism of action. Molecular docking of **O₄SE₆** and **O₄RE₆** to the DRD2/haloperidol and DRD2/risperidone crystal structures showed that only DRD2/haloperidol crystal structure could recapture the interaction between Phe110$^{3.28}$ and RHS cyclohexane moiety (Supplementary Fig. 7a, b). This is a direct consequence of the conformational rearrangements in DRD2— the relocation of Trp100$^{EL1}$, which consequently affects the formation of the SEBP at DRD2, allowing the cyclohexyl substituents of **O₄RE₆** and **O₄SE₆** to form a hydrophobic contact with the benzene ring of Phe110$^{3.28}$ in TM3 (Supplementary Fig. 7a, b). Further experimental data also supported the docking poses, due to the similar behaviors between **O₄SE₆** and **O₄RE₆**, including Gα$_{i/o}$ and β-arrestin2 agonism activity at DRD2 or its Phe110$^{3.28}$ mutants (Fig. 4a–c and Supplementary Tables 5, 6).

**Structure-based optimization towards selective DRD2 agonists**. The docking poses of **O₄SE₆** and **O₄RE₆** at DRD2 showed that these ligands also adopted the same binding pose as haloperidol or risperidone in the complex (Supplementary Fig. 7a, b and Fig. 2e, f). Even though the molecular sizes of **O₄SE₆** and **O₄RE₆** are obviously smaller than those of haloperidol and risperidone, but more comparable to the size of eticlopride and nemonapride, they failed to bind the DRD2 in an orientation analogous to those of the latter two in the DRD3 and DRD4 structures, respectively (Fig. 2e–h). This is likely a direct consequence of the inward shift of TM5 at DRD2 (DRD2 vs. DRD3/DRD4) (Supplementary Fig. 5), which consequently shrinks the OBP substantially so that it would preclude a shallow ligand-binding mode at DRD2 (Supplementary Fig. 5), allowing **O₄SE₆** and **O₄RE₆** to engage a deep binding pose (Supplementary Fig. 7a, b).

The conserved serine residues on TM5 (5.42, 5.43, and 5.46) have been previously reported to form the structural basis of agonist and partial agonist actions at β₁ and β₂ adrenergic receptors (AR)[24–26]. These conserved serine residues of DRD2 are also attributable to ligand efficacy and overall G-protein activation[27–30]. In the docked poses of **O₄SE₆** or **O₄RE₆** at DRD2, the benzofuran moiety also interacts with the conserved serine residues at DRD2 (Supplementary Fig. 7a, b), consistent with their agonist activity at DRD2 (Fig. 4b, c and Supplementary Tables 5, 6). And, the substitutions of these conserved serine residues with glycine impaired **O₄SE₆**'s and **O₄RE₆**'s agonism (Supplementary Fig. 8), without altering DRD2 expression levels (Supplementary Fig. 6c). Unexpectedly, when we compared the G-protein agonist activity of **O₄SE₆** and **O₄RE₆** with **O₄LE₆** at DRD2 and DRD3, we observed that **O₄SE₆** showed G-protein signaling agonist activity at DRD2 (Gα$_{i/o}$ agonism EC$_{50}$ = 18.45 nM) and no detectable agonist activity at DRD3; and, **O₄RE₆** displayed agonist activity at both receptors but had a lower efficacy at DRD3 (Gα$_{i/o}$ signaling $E_{max}$ = 94.46% for DRD2/ 52.44% for DRD3) (Fig. 4d, f and Supplementary Table 5).

The comparison of G-protein-signaling action across D₂-like receptor subtypes is challenging, since DRD2 promiscuously couples to all members of the Gα$_{i/o}$ family of G proteins, whereas the DRD3 selectively couples to the Gα$_o$ subunit[31,32]. Alternatively, the measurement of G-protein independent β-arrestin recruitment provides a feasible way, since all D₂-like receptor subtypes can induce β-arrestin translocation[33,34]. Then, β-arrestin2 recruitment assay was applied to the **O₄SE₆** and **O₄RE₆** at D₂-like receptors. The similar results recapitulated our findings obtained from measuring Gα$_{i/o}$-mediated cAMP-inhibition activity. **O₄SE₆** showed agonist activity at DRD2 (β-arrestin2 recruitment EC$_{50}$ = 1055 nM) and no detectable agonist activity at DRD3; and, **O₄RE₆** displayed different efficacy agonist activity at both receptors (β-arrestin2 recruitment $E_{max}$ = 99.61% for DRD2/21.41% for DRD3) (Fig. 4e, g and Supplementary Table 6). Compared to the deep binding poses of **O₄SE₆** and **O₄RE₆** at DRD2 (Supplementary Fig. 7a, b), molecular docking of **O₄SE₆** and **O₄RE₆** to the DRD3 crystal structure showed that they adopted similar shallow binding poses as that of eticlopride in the complex (Supplementary Fig. 7c, d). These results indicate that the orientation of the ligand at the OBP and its interaction with TM5 could be another key factor to facilitate DRD2 versus DRD3 functional selectivity.

To obtain further insights into the orientation of different OBP-binding moieties at DRD2, we used the initial hit **23991615** (**O₉LE₉**) as a template and synthesized an analog—**O₇LE₆** (Fig. 5a). While retaining the LHS phenyl group for binding at DRD2 OBP, a (1-hydroxycyclohexyl)methyl substitution was introduced in **O₇LE₆** in replacement of the butyl group in **O₉LE₉**, to facilitate interaction with Phe110$^{3.28}$. Molecular docking suggested that compound **O₇LE₆** adopted different binding poses

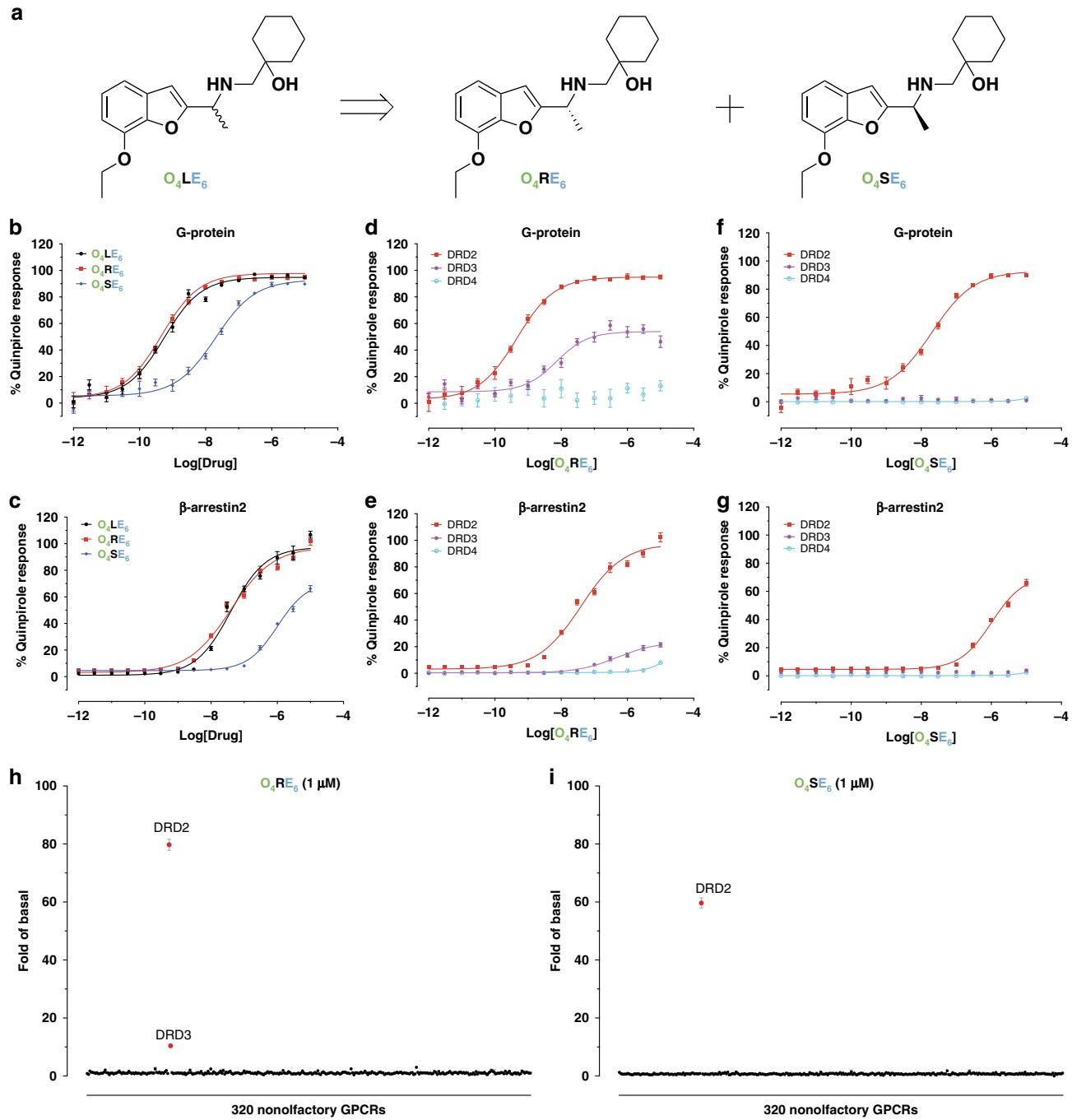

**Fig. 4 Comparison of functional activity of individual enantiomers—O₄RE₆ and O₄SE₆. a** The chemical structures of **O₄LE₆**, **O₄RE₆**, and **O₄SE₆**. **b**, **c** Stereoisomers of **O₄LE₆** in DRD2-mediated G protein activity (G$_{\alpha i/o}$-mediated cAMP inhibition; **b**) and β-arrestin2 recruitment (Tango; **c**). **d**, **e O₄RE₆** is a DRD2 full agonist and DRD3 partial agonist in G protein activity **d** and β-arrestin2 recruitment **e**. **f**, **g O₄SE₆** is a selective DRD2 agonist in G protein activity **f** and β-arrestin2 recruitment **g**. Data normalized to percent quinpirole activity. All data represent three independent experiments performed in triplicate technical replicates and in parallel using the same drug dilutions. Error bars, SEM ($n = 3$ independent experiments). See also Supplementary Tables 5 and 6. **h**, **i O₄RE₆** (**h**) and **O₄SE₆** (**i**) at 1 μM were screened against 320 non-olfactory GPCRs for agonism in the β-arrestin2 recruitment Tango assay. Each point shows luminescence normalized to basal level at a given GPCR. Data are mean ± SEM of non-normalized results ($n = 4$). Source data are provided as a Source Data file.

at DRD2 and DRD3 (deep vs. shallow binding pose, respectively). And, the 2-nitro substituent of **O₇LE₆** oriented to different directions at DRD2 and DRD3 (Ser[5.42] of TM5 for DRD2 versus Ser[5.46] of TM5 for DRD3) (Supplementary Fig. 7e–h). Compared to the flexible ethoxy substituent at **O₄LE₆**, the rigid 2-nitro substituent at **O₇LE₆** recaptured interaction with TM5 at DRD3 (Supplementary Fig. 7e–h). In functional assays, **O₇LE₆** is a

DRD2 and DRD3 agonist in both G$\alpha_{i/o}$ signaling (DRD2 EC$_{50}$ = 1.13 nM, $E_{max}$ = 99.98%; DRD3 EC$_{50}$ = 1.96 nM, $E_{max}$ = 82.68%) and β-arrestin2-recruitment assays (DRD2 EC$_{50}$ = 6.95 nM, $E_{max}$ = 85.33%; DRD3 EC$_{50}$ = 117.1 nM, $E_{max}$ = 88.46%) and no DRD4 activity (Fig. 5b, c and Supplementary Tables 5, 6). And, the Ser197[5.46]Gly substitution substantially diminished **O₇LE₆**'s agonism at DRD2 (Supplementary Fig. 9a).

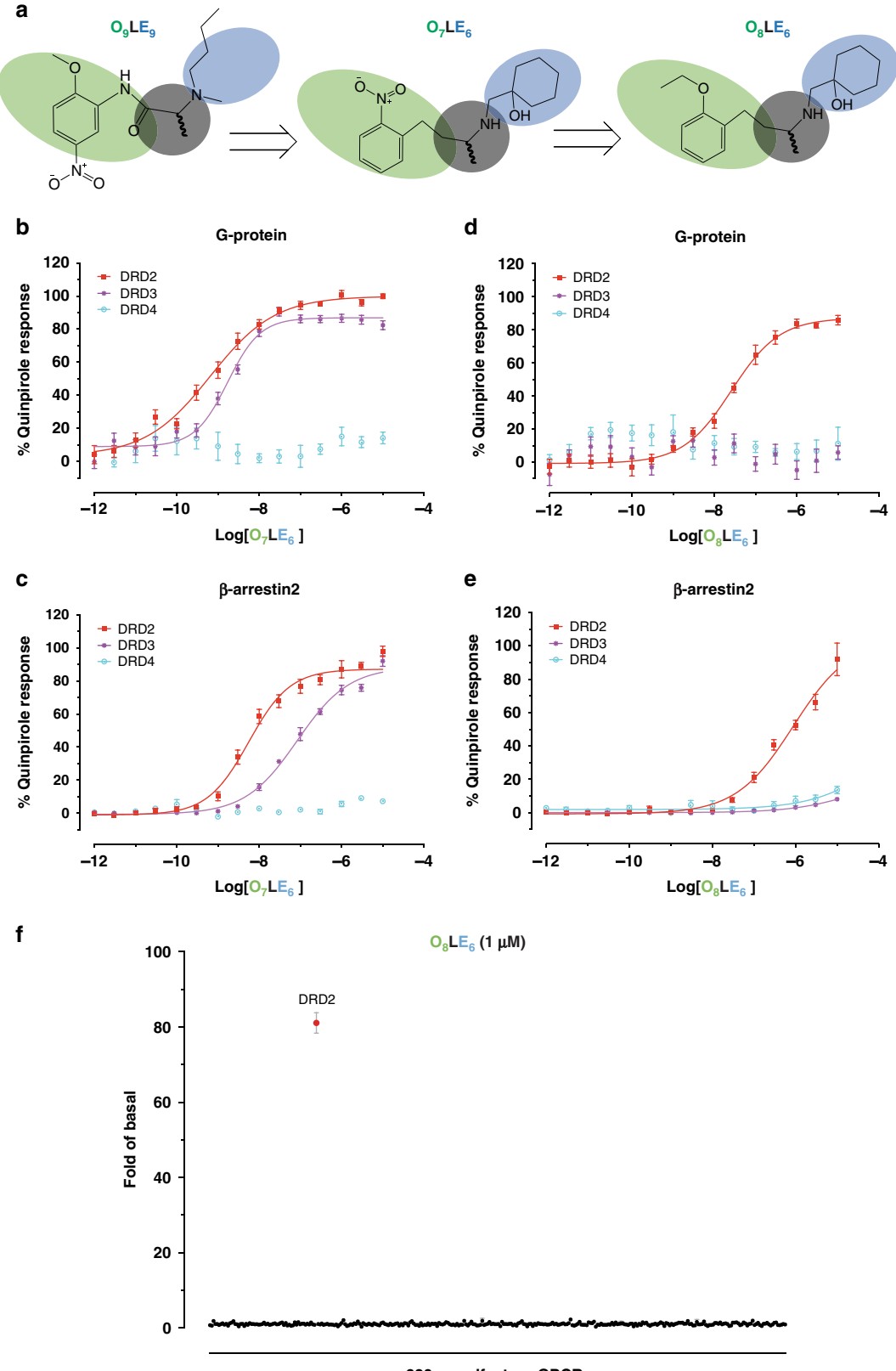

**Fig. 5 Design of DRD2 selective agonist and validation of its functional activity. a** The overview of structure-guided analogs of **O₉LE₉**. **b–e** Normalized concentration-response studies for analogs of **O₉LE₉** in D₂-like receptors mediated activation of Gα$_{i/o}$ (Gα$_{i/o}$-mediated cAMP inhibition; **b**, **d**) and β-arrestin2 translocation (Tango; **c**, **e**), normalized to percent quinpirole activity. Data represent three independent experiments performed in triplicate technical replicates and in parallel using the same drug dilutions. Error bars, SEM ($n = 3$ independent experiments). See also Supplementary Tables 5 and 6. **f** **O₈LE₆** at 1 μM was screened against 320 non-olfactory GPCRs for agonism in the β-arrestin2 recruitment Tango assay. Each point shows luminescence normalized to basal level at a given GPCR. Data are mean ± SEM of non-normalized results ($n = 4$). Source data are provided as a Source Data file.

To further test the hypothesis that ligand contacts with the Phe$^{3.28}$ of the SEBP and the conserved Ser$^{5.42/5.43/5.46}$ of the OBP facilitates DRD2 versus DRD3/DRD4 functional selectivity, we synthesized another analog—$O_8LE_6$, in which the same flexible ethoxy substituent as in $O_4LE_6$ was attached to position 2 of the benzene ring to replace the rigid 2-nitro substituent at $O_7LE_6$ (Fig. 5a and Supplementary Fig. 7i, j). Based on the different docking poses of $O_7LE_6$ at DRD2 and DRD3 (Supplementary Fig. 7e–h), the flexible ethoxy substituent, just like the one in $O_4LE_6$, may disrupt the agonism activity at DRD3. As predicted, $O_8LE_6$ displayed agonist activity at DRD2 ($G\alpha_{i/o}$ agonism $EC_{50}$ = 30.42 nM and β-arrestin2 recruitment $EC_{50}$ = 311.0 nM) and no detectable agonist activity at DRD3 and DRD4 (Fig. 5d, e and Supplementary Tables 5, 6). With the substitution of conserved serine residues with glycine, $O_8LE_6$ lost its potency and efficacy in both G-protein and β-arrestin2 assays (Supplementary Fig. 9b). And, we also confirmed $O_8LE_6$ has no detectable antagonist activity at DRD3 and DRD4 (Supplementary Fig. 10a–c).

A major goal of this study was to prove DRD2 activation mechanism via both the OBP and the SEBP, which could be suitable for the design of subtype-selective DRD2 agonists. To investigate compound specificity more broadly, $O_4RE_6$, $O_4SE_6$, and $O_8LE_6$ were then counter-screened for agonism against 320 nonolfactory GPCRs via β-arrestin2 recruitment Tango assay[34]. $O_4RE_6$ showed agonist efficacy at DRD2 and DRD3 at 1 μM (the efficacy at DRD3 is much lower than that at DRD2), whereas only DRD2 activity was observed for $O_4SE_6$ or $O_8LE_6$ at 1 μM concentration (Figs. 4h, i and 5f). It would be impracticable to check antagonist activity of $O_4SE_6$ or $O_8LE_6$ for each receptor, but we confirmed that there is no detectable antagonist activity at 12 serotonin receptors and 5 dopamine receptors (Supplementary Fig. 10). Beside agonist activity at DRD2, $O_4SE_6$, $O_4RE_6$, and $O_8LE_6$ activate $5HT_{1D}$ and $5HT_{7A}$ receptors at high concentrations (over 1 μM). These results confirmed our hypothesis that highly subtype-selective DRD2 ligands could be identified through an integrative approach combining structure-based and mechanism-driven screening and lead optimizations.

## Discussion

The discovery of highly selective DRD2 ligands has been extremely challenging due to the high similarity of its ligand-binding pocket to those of DRD3 and DRD4. But on the other hand, since the distance between EBP and OBP is longer in DRD3 than in DRD2 (SEBP-OBP) and DRD4 (EBP-OBP)[16,35], highly DRD3-selective compounds, such as R-22[16,35], can be obtained by designing compounds with a longer linker between the OBP-binding moiety and EBP-binding moiety. For DRD4, which has a specifically larger EBP adjacent to the OBP, highly selective ligands have also been reported[13,14]. As for DRD2, the previous identified EBP[15] is an unsealed pocket which is not the ideal druggable pocket for SBDD which relies on rigid pockets[17]; and the identified SEBP in the DRD2/haloperidol structure is also a flexible pocket, not suitable for SBDD either. Furthermore, the SEBP is extremely small in size compared to the EBP of DRD4, and closer to the OBP in distance than the DRD3 EBP. This structural information may explain why there is no DRD2-selective ligand so far, and all known DRD2-targeted drugs (such as all FDA-approved antipsychotics) concomitantly bind DRD3 and DRD4.

Here leveraged by the DRD2/haloperidol crystal structure, we present a combination of structural, computational, and pharmacological studies that illuminate the structure and function of the DRD2 SEBP. We explored the previously unrecognized mechanism of DRD2 activation via the SEBP, thereby illuminating the different binding pose (shallow vs. deep) and activation

mechanism could be as a key controller to distinguish DRD2 versus DRD3/DRD4 selectivity. Through this approach, we rapidly discovered two highly selective DRD2 agonists. Methodologically, the combination of structure-based design and mechanism-driven screening may have a broader application in accelerating the discovery of selective ligands to distinguish extremely similar receptors, which are still a large portion of drug targets.

## Methods

**Expression and purification of DRD2.** Constructs encoding DRD2 for the generation of crystals were based on previously published DRD2 constructs in which T4L residues 2–161[15]—was fused into third intracellular loop of DRD2 (V223–R361) with truncations of the N termini residues 1–34 and three thermo-stabilization mutations I122$^{3.40}$A, L375$^{6.37}$A, and L379$^{6.41}$A. The modified DRD2-T4L protein was expressed in *Spodoptera frugiperda* (Sf9) cells (Expression Systems) using Bac-to-Bac Baculovirus Expression System (Invitrogen) for 48 h. The insect cells were disrupted by repeated washing and centrifugation, with hypotonic buffer (10 mM HEPES, 10 mM MgCl$_2$, 20 mM KCl, pH 7.5) containing protease inhibitors (500 μM AEBSF, 1 μM E-64, 1 μM leupeptin, 150 nM aprotinin) (one time) and high-osmotic buffer (1.0 M NaCl, 10 mM HEPES, pH 7.5, 10 mM MgCl$_2$, 20 mM KCl) (three times). Purified membranes were resuspended in a buffer containing 10 mM HEPES, pH 7.5, 10 mM MgCl$_2$, 20 mM KCl, 150 mM NaCl, 20 μM haloperidol (sigma), and protease inhibitors cocktail (roche), and incubated at room temperature for 1 h. After a 30 min incubation at 4 °C in the presence of 2 mg/mL iodoacetamide (Sigma), membranes were solubilized in 10 mM HEPES, 150 mM NaCl, 1% (wt/vol) n-dodecyl-β-D-maltopyranoside (DDM, Anatrace), 0.2% (wt/vol) cholesteryl hemisuccinate (CHS, Sigma) for 2 h at 4 °C. Unsolubilized material was removed by centrifugation at 150,000×g for 30 min, followed by incubation in 20 mM buffered imidazole (pH 7.5), 800 mM NaCl with TALON IMAC resin (Clontech) at 4 °C, overnight. The resin was then washed with 10 column volumes (CVs) of Wash Buffer I (50 mM HEPES, pH 7.5, 800 mM NaCl, 0.1% (w/v) DDM, 0.02% (w/v) CHS, 20 mM imidazole, 10% (v/v) glycerol, and 10 μM haloperidol), followed by 10 CVs of Wash Buffer II (25 mM HEPES, pH 7.5, 150 mM NaCl, 0.05% (w/v) DDM, 0.01% (w/v) CHS, 10% (v/v) glycerol, and 10 μM haloperidol). The protein was then eluted in 3–4 CVs of Elution Buffer (50 mM HEPES (pH 7.5), 50 μM haloperidol, 500 mM NaCl, 10% (v/v) glycerol, 0.05% (w/v) DDM, 0.01% (w/v) CHS, and 250 mM imidazole). Imidazole was removed by desalting the protein over PD MiniTrap G-25 columns (GE Healthcare). The protein was then treated overnight with His-tagged TEV protease (homemade) and His-tagged PNGase F (NEB) to remove the N-terminal His-tag, Flag-tag and deglycosylate the receptor. His-tagged TEV protease, His-tagged PNGase F, cleaved His-tag and uncleaved protein were removed by passing the suspension through equilibrated TALON IMAC resin (Clontech) and collecting the flowthrough. The DRD2/haloperidol complexes were then concentrated to ~40 mg/mL with a 100 kDa molecular mass cut-off Vivaspin 500 centrifuge concentrator (Sartorius Stedim). Protein purity and monodispersity were tested by analytical size-exclusion chromatography.

**Lipidic cubic phase (LCP) crystallization.** DRD2/haloperidol complexes were reconstituted into the LCP by mixing protein and a monoolein:cholesterol mixture at a ratio of 40%:54%:6% by using the twin-syringe method[36]. Crystallization was performed on 96-well glass sandwich plates using a handheld dispenser (Art Robbins Instruments), dispensing 45 nL of protein-laden LCP and 1 μl precipitant solution per well. Plates were then incubated at 20 °C. Crystals were obtained in 100 mM Tris/HCl pH 7.5, 150 mM sodium malonate, 30% PEG400, and grew to full size around 1 week. The crystals were harvested directly from the LCP matrix using micromount (MiTeGen) and flash frozen in liquid nitrogen.

**Data collection and structure determination.** X-ray diffraction data of DRD2/haloperidol crystals were collected at Spring-8 beam line 41XU, Hyogo, Japan, using a PILATUS detector (Proposal Number: 2019B2715), and GM/CA at APS of Argonne National Lab, using Eiger 6M detector. The crystals were exposed to 0.5 s of unattenuated beam using 0.5° oscillation per frame. Diffraction images of six crystals were indexed, integrated, and scaled using HKL3000[37]. Initial phase information was obtained by molecular replacement (MR) with the program PHASER[38] using two independent search models—a receptor portion of the DRD2/risperidone complex (PDB code: 6CM4), and the T4L portions of β2AR-T4L (PDB code: 2RH1) as initial models. Refinement was performed with PHENIX[39] and REFMAC followed by manual examination and rebuilding of the refined coordinates in the program COOT[40] using $|2F_o|−|F_c|$, $|F_o|−|F_c|$, and omit maps. After the refinement, the real space correlation coefficient (RSCC) value of the haloperidol is 0.94, which means the electron density is proper fitting of the ligand haloperidol.

**Radioligand-binding assay.** Binding assays were performed using HEK293 T (ATCC CRL-11268; mycoplasma free) membrane preparations transiently

expressing DRD2 ($D_2$ long receptor, pcDNA3.1), DRD3 (pcDNA3.1), DRD4 ($D_{4.4}$ variant, pcDNA3.1), or different mutants. HEK293 T cells (ATCC CRL-11268; mycoplasma free) were transfected (PEI transfection) and membrane preparation and radioligand-binding assays were set up in 96-well plates[41]. Binding assays were conducted in 96-well in standard-binding buffer (50 mM Tris, 10 mM MgCl$_2$, 0.1 mM EDTA, 0.1% BSA, pH 7.4) using [$^3$H]-N-methylspiperone (PerkinElmer) as the radioligand. For displacement experiments, increasing concentrations of compounds were incubated with membrane and radioligands (0.8–1.0 nM [$^3$H]-N-methylspiperone) for 2 h at room temperature in the dark. Saturation binding assays with 0.01–15 nM [$^3$H]-N-methylspiperone in standard binding buffer were performed to determine equilibrium dissociation constant ($K_d$) and $B_{max}$, whereas 10 µM final concentration of haloperidol was used to define nonspecific binding. The reaction was terminated by rapid vacuum filtration onto chilled 0.3% PEI-soaked GF/A filters followed by three quick washes with cold washing buffer (50 mM Tris–HCl, pH 7.4). Data were analyzed with GraphPad Prism 6.0 using 'One-site-homologous' to yield $K_d$, 'One-site-Fit $K_i$' to yield $K_i$.

**Split-luciferase-based cAMP reporter assays.** HEK293T (ATCC CRL-11268; mycoplasma free) cells co-expressing DRD2 ($D_2$ long receptor, pcDNA3.1), DRD3 (pcDNA3.1), DRD4 ($D_{4.4}$ variant, pcDNA3.1), or different mutants along with a split-luciferase-based cAMP biosensor (GloSensor; Promega) were seeded in 384-well white clear bottom cell culture plates (Corning; 10,000 cells/well, 40 µL/well) in DMEM containing 1% dialyzed FBS (Omega Scientific). The next day, culture medium was removed and 20 µL/well of drug buffer was added followed by addition of 10 µL of 3 × drug solutions for 15 min at room temperature. To measure agonist activity for $G\alpha_{i/o}$-coupled receptors, 10 µL luciferin (4 mM final concentration) supplemented with isoproterenol (400 nM final concentration was added to activate Gs via endogenous $\beta_2$-adrenergic receptors) and luminescence intensity was quantified 15 min later. Data were analyzed using "log(agonist) vs. response" in GraphPad Prism 6.0. Data were normalized to percent quinpirole response, which was present in every experiment.

**Tango arrestin recruitment assay.** Tango constructs ($5HT_{1A/1B/1D/1E/1F/2A/2B/2C/4/5A/6/7A}$, DRD1, DRD2: D2 long receptor, DRD3, DRD4: $D_{4.4}$ variant and DRD5) were designed and assays were performed as previously described[34]. Briefly, HTLA cells expressing the TEV fused-$\beta$-arrestin2 (kindly provided by Dr. Bryan L. Roth) were transfected (PEI) with serotonin receptors, dopamine receptors or different mutants. Next day, cells were plated into white 384-well white clear bottom cell culture plates (Corning; 10,000 cells/well, 40 µL/well) in DMEM containing 1% dialyzed FBS (Omega Scientific). The following day, drug solutions were prepared in drug buffer (1 × HBSS, 20 mM HEPES, 0.1% BSA, 0.01% ascorbic acid, pH 7.4) at 3 × final concentration and added to the cells (20 µL/well) for overnight incubation. The next day, media was decanted and replaced with 20 µL/well of Bright-Glo reagent (Promega, after 1:20 dilution). After 20 min, plates were read on a Envision (Perkin Elmer) at 1 s per well. Data were analyzed using "log(agonist) vs. response" in GraphPad Prism 6.0. Data were normalized to percent quinpirole response, which was present in every experiment.

**GPCRome screening.** Agonist activity at 320 non-olfactory GPCRs ("human GPCRome") was based on Tango Arrestin Recruitment Assay with modifications as indicated below[19,34]. Briefly, HTLA cells were plated in 384-well white clear bottom plates in DMEM supplemented with 10% FBS (10,000 cells in 40 µL/well). After overnight incubation, cells replaced with 40 µL/well of fresh DMEM supplemented with 2% FBS and transfected (PEI) with receptor DNA (20 ng/well) for 24 h. Medium was removed and replaced with 40 µL/well of DMEM supplemented with 1% dialyzed FBS, followed by 10 µL/well drug solution at 5× of a final concentration (1 µM). Medium with 1% dialyzed FBS served as a baseline response for each receptor. After overnight incubation (~18 h), medium and drug solutions were removed and 20 µL/well of BrightGlo reagents (Promega) were added. Luminescence (Relative Luminescence Unit, RLU) was read on a luminescence reader, Envision (Perkin Elmer), after 20 min incubation at RT. The assay was designed so that 40 receptors were tested in each 384-well plate; each receptor was stimulated in four replicate wells with drug and four replicate wells with 1% dialyzed FBS as a control. DRD2 served as an assay control—16 replicate wells with 0.1 µM Quinpirole and 16 replicate wells with 1% dialyzed FBS. Additional 32 wells served as background control. The GPCRome was accordingly screened in a total of eight 384-well plates. Results were presented in the form of fold of basal for each receptor and plotted in GraphPad Prism 6.0.

**Bioluminescence resonance energy transfer (BRET) assay.** To measure DRD2-mediated G protein activation, HEK293T cells (ATCC CRL-11268; mycoplasma free) were co-transfected (PEI) with human DRD2 ($D_2$ long receptor, pcDNA3.1), $G\alpha_{i1}$ containing C-terminal *Renilla* luciferase (RLuc8, pcDNA3.1), G$\beta$ and G$\gamma_2$ containing a C-terminal GFP (pcDNA3.1, at mass ratio 1:1:1:1, respectively). After at least 16 h, transfected cells were plated in poly-lysine coated 96-well white clear bottom cell culture plates in plating media (DMEM + 1% dialyzed FBS) at a density of 40–50,000 cells in 200 µL/well and incubated overnight. The next day, media was decanted, and cells were washed twice with 60 µL of drug buffer (20 mM HEPES, 1X HBSS, pH 7.4), then 60 µL of the RLuc substrate, coelenterazine 400a (Promega, 5 µM final

concentration in drug buffer), was added per well, incubated an additional 5 min to allow for substrate diffusion. Afterwards, 30 µL of drug (3×) in drug buffer was added per well and incubated for another 5 min. Plates were immediately read for luminescence at 400 nm and GFP fluorescent emission at 515 nm for 1 s per well using a Mithras LB940 multimode microplate reader. The ratio of GFP/RLuc was calculated per well and the net BRET ratio was calculated by subtracting the GFP/RLuc from the same ratio in wells without GFP present. The net BRET ratio was plotted as a function of drug concentration using Graphpad Prism 6.0.

**Molecular docking.** The compounds were docked to the DRD2/haloperidol, DRD2/risperidone (PDB: 6CM4) and DRD3/eticlopride (PDB: 3PBL) crystal structures using the open source software Autodock Vina 1.1.2[42]. The resulting docked compound poses were scored by summing the receptor-ligand electrostatics, van der Waals interaction energies and corrected for context-dependent ligand desolvation. The receptors were prepared by adding hydrogens, charges, and repairing missing atoms. The compounds were drawn in ChemBioDraw Ultra 12.0 followed by MM2 minimization of ligands by keeping a check on the connection error in the bonds. Ligands and Grid preparation was done using AutoDock Vina 1.1.2[42] in order to carry out molecular docking analysis.

**General chemistry procedures.** The reaction conditions and yields were not optimized. All commercial chemicals and solvents were used as obtained without further purification. Microwave reactions were run in a Biotage Initiator microwave reactor. Synthetic intermediates were purified on 230−400 mesh silica gel on a Teledyne CombiFlash $R_f$ flash chromatography. $^1$H NMR spectra were recorded on Bruker AVANCE-II or AVANCE-III spectrometers at 600 or 800 MHz. $^{13}$C NMR spectra were recorded on AVANCE-III spectrometer at 200 MHz. NMR chemical shifts were reported in $\delta$ (ppm) using residual solvent peaks as standards (CDCl$_3$–7.26 (H); CD$_3$OD–3.31 (H), 49.00 (C)). Mass spectra were measured using an LCMS-IT-TOF (Shimadzu) mass spectrometer in ESI mode. The purity of all final compounds (>95%) was determined by analytical HPLC (Shim-pack GIST C$_{18}$ column (250 × 4.6 mm, particle size 5 µM); 0.05% TFA in H$_2$O/0.05% TFA in MeOH gradient eluting system; flow rate = 1.0 mL/min, $\lambda$ = 254 or 280 nm). Synthetic procedures for OLE compounds can be found in Supplementary Note 2.

**Reporting summary.** Further information on research design is available in the Nature Research Reporting Summary linked to this article.

## Data availability
Data supporting the findings of this manuscript are available from the corresponding authors upon reasonable request. A reporting summary for this Article is available as a Supplementary Information file. The source data underlying Figs. 1a, 3, 4b–i, 5b–f and Supplementary Figs. 6, 8–10, 11b–e are provided as a Source Data file. Atomic coordinates and structure factor files for the DRD2/haloperidol structure have been deposited in the RCSB Protein Data Bank with identification code 6LUQ.

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

## Acknowledgements

This work was supported by Strategic Priority Research Program of the Chinese Academy of Sciences (XDB19000000) to S.W.; the Thousand Talents Plan-Youth to S.W.; the Natural Science Foundation of Shanghai (19ZR1466200) to S.W.; the Natural Science Foundation of China (81703361) to J.C.; and by Shanghai Municipal Government and ShanghaiTech University to J.C. We gratefully acknowledge the staff of BL41XU@Spring-8, BL18U1, and BL17U @National Center for Protein Sciences Shanghai and GM/CA@Advanced Photon Source.

## Author contributions

L.F. performed pharmacological assays, analyzed the data, and assisted with preparing the manuscript; L.T. synthesized analogs and performed analytical chemical analysis; Z.C. assisted with protein expression, purified, and crystallized the receptor, optimized crystallization conditions, grew crystals for data collection; F.N. performed GPCR-ome assay and analyzed the data; J.Q. assisted with protein expression, purified, and crystallized the receptor; Z.L. provided independent structure quality control analysis and assisted with structure determination; J.C. supervised ligand synthesis and edited the manuscript; S.W. was responsible for the overall project strategy, project management, solving and analyzing the structure, and the writing of the manuscript.

## Competing interests

The authors declare no competing interests.
