## [Peer Review File · Nature Communications]

Reviewers' Comments:

Reviewer #1:

Remarks to the Author:

In their manuscript entitled "Atypical structure-based discovery of selective D2 dopamine receptor ligands" by Luyu, Fan, Liang Tan, Zhangcheng Chen, Jianzhong Qi, Fen Nie, Zhipu Luo, Jianjun Cheng, and Sheng Wang, the authors address a topic that is very important to the field, the discovery and development of subtype-selective Dopamine D2 receptor (DRD2) ligands. One of the most important findings of the paper is a new X-ray crystal structure of the DRD2 in complex with the clinical antipsychotic haloperidol. This structure will be definitely appreciated by the community, as it has the potential to provide new insights into ligand binding at DRD2. However, it seems that this relevant result is over-shadowed by the attempt to develop novel DRD2 selective ligands employing a very unclear rationale. Even the title of the manuscript does not provide any hint to the new crystal-structure and this might lead to the publication being unnoticed by certain researchers in this field (like structural biologists, computer chemists etc.). Overall, the manuscript is not easy to read as it contains a lot of over-detailed description on data on one side, while on the other hand it is clearly lacking a good description of the rationale for the individual experiments performed. In its present form, the manuscript is thus not suitable for publication. A number of the logical and technical shortcomings are detailed below.

1) Unclear or rationale: The authors start with a brief description of their new crystal structure of DRD2 in complex with haloperidol which they refer to as DRD2-preferring. However, based on the binding data from Fig 1a it might not be justified. Furthermore, similar affinities to all D2-like receptors are demonstrated in IUPHAR/BPS Guide to Pharmacology. It is unclear what they intend to learn from this structure with a rather unselective ligand in the context of D2-selective ligands. Moreover, the differences to the previously published structure with risperidone are only sparsely discussed. How is it concluded, that DRD2 has a variable extended binding pocket (EBP) while those of DRD3 and DRD4 are considered to be rigid? Is this a new finding of the present study?

2) Can the authors explain why the extracellular sides of TM1 and TM2 in DRD2-haloperidol are shifted compared to the structure of DRD2-risperidone? Could it be just a crystallization artefact?

3) Absence of actual structure-based design: instead of virtually screening the extended binding pocket of any of the DRD2 crystal structures for the discovery of selective ligands, the authors used ligands from a previous study with DRD4 that were characterized as high-ranking or mid-ranking molecules for DRD4. It remains unclear how this can be considered as a structure-inspired discovery of DRD2-selective ligands. The additional insight gained by obtaining the crystal structure of DRD2-haloperidol does not seem to be related to this part of the manuscript. It is completely unclear how "this approach may have a broader application in accelerating the discovery of selective ligands" as claimed in the discussion.

4) The authors claim to have "explored a previously unrecognized mechanism of DRD2 activation via the VBP" and suggest that the different binding pose (shallow vs deep) could be a key controller of DRD2 selectivity. From the present data, this conclusion does not appear to be fully justified, as the "shallow vs. deep" comparison between DRD2 and DRD3/4 includes ligands with very different molecular structure and size (benzamides nemonapride and eticlopride) vs. haloperidol/risperidone. Moreover, these crystal structures were obtained with antagonists, and it is therefore hard to conclude whether the same differences would be obtained with agonists.

5) The conclusions drawn from the SAR studies are peculiar, if not false. For example, the authors claim "For the 6 analogs (O1-6LE6) with various substitutions on the LHS benzofuran, the most potent analogs are O4 and O5 compounds, which have ethoxy and propoxy substituents respectively." From the binding affinities shown in Supplementary table 4, it rather looks like there is no significant difference between O4, O5 and O6 and no correlation between the length of the alkyl chain and determined affinities. Also, the statement "For the RHS cyclohexane part (O4LE1-9), we found that the best substituents are cyclohexane (E6) and cycloheptane (E7)" is not accurate since the best results were obtained with O4LE8 and larger rings were not synthesized.

6) It is astonishing that more than a 1000-fold difference in affinity ($K_i = 1.9 \mu\text{M}$ for O4LE6 to DRD2) and potency ($\text{EC}_{50} = 0.57 \text{ nM}$) is obtained. This should be addressed thoroughly in the results section of the manuscript.

7) The authors claim that their work yielded highly selective DRD2 agonists, compounds O4SE6 and O8LE6. This key finding is only supported by functional assays and surprisingly ligand binding assays to determine the affinities of these compounds were not performed. Moreover, selectivity was only investigated in the TANGO- β -arrestin assay using agonist mode, neglecting the fact that these compounds could act as antagonists of other receptors.

8) Misleading abstract and introduction: From the abstract and the introduction the reader gets the impression that non subtype-selective dopamine receptor antagonists cause life-threatening side effects by interaction with DRD3 or DRD4, while the references indicated actually refer e.g. to valvular heart disease caused by 5-HT_{2B} receptors.

9) Wrong referencing to Figures and Tables throughout the text: on various occasions, the authors describe results and reference to Figures that actually do not contain this data. E.g., "The docking pose of O4SE6 and O4RE6 at DRD2 showed that these ligands also adopted the same binding pose as haloperidol or risperidone in the complex (Figure S8a/b)." Instead, Figure S8 compares DRD2-haloperidol and DRD3-eticlopride.

10) Use of tables: the numerous different tables containing very similar results are confusing to the reader e.g. Tables S4 and S7 both contain cAMP inhibition signaling and could be combined. Sometimes, the same data is presented in one table even more than once, e.g. O4LE6 in Table S4.

11) Technical issues

- The chemical procedures lack general information on the purchase of reagents, instruments used etc.
- No information on test compound purity is provided. This is critical when discussing SAR.
- The experimental methods describing the synthesis of test compounds are lacking sufficient details allowing repetition of the procedure. For each compound at least the exact amount of reagents (mmols, equivalents) and solvents as well as the yields should be given.
- The nomenclature of the compounds O4LE6 vs O5LE6 etc. is confusing, as the reader needs to pay very detailed attention to the individual numbers
- The description of the biological data is partly lacking important details, e.g. how were the cells transfected (reagent, cDNA vectors etc.)
- The composition of all the assay buffers and the origin of commercially available reagents should be specified. For example, where does the His-tagged TEV protease come from?
- Comparison of the expression levels of mutated receptors: the authors conclude similar expression levels from receptor constructs that were largely modified on their C-termini (additional expression of RLuc). Moreover, the authors directly measure RLuc activity after addition of the Coelenterazine substrate. Usually, an incubation time is needed to assure that Coelenterazine is able to cross the cell membrane, as the receptor C-terminus is and thus the luciferase is located intracellularly. The authors should use saturation binding to compare the expression of the individual receptors.
- While presenting binding curves or functional assays it should be clearly stated whether the displayed graph is obtained from a pooled data or is shown as a representative graph from a single experiment (Fig. 1a etc.).
- Representation of the amino acid mutations should be carefully thought through. At the moment many different variants are used for one position (like F3.28A, F1103.28A, Phe1103.28Ala,). This seems to be also confusing the authors, since sometimes the mutations are referred to in the text incorrectly.
- The authors are encouraged to think about how many significant figures are appropriate for measurement results. For consistency of reporting, the same number of significant figures should be used within an assay (at the moment it varies from 1 to 6).
- There are numerous grammatical and spelling errors throughout the text.

Reviewer #2:

Remarks to the Author:

The manuscript "Atypical structure-based discovery of selective D2 dopamine receptor ligands" by Sheng Wang and colleagues has very interesting results and could potentially promote new

development of dopamine receptor drugs. The authors investigated the D2 dopamine receptor (DRD2), which is one of the most important therapeutic targets in neuropsychiatric diseases. No real DRD2-selective drugs are available yet. The authors showed a new (so-called variable part or VBP) extension of the orthosteric binding pocket at DRD2, which differs from other D2-like subtypes (DRD3 or DRD4) and can be used for the optimization of new ligands, based on the crystal structure of a DRD2-haloperidol complex. Using a structure-based design approach, they found highly selective agonists for DRD2, which could play an important role for highly specific binding and downstream signaling in the future.

The manuscript unfortunately reads very technically in many parts and is very descriptive, which is certainly due to the large number of compound descriptions.

The methods (assays and X-ray crystallography) used for investigation are state-of-the-art and of high quality.

I cannot evaluate the chemical synthesis of these compounds.

I believe the manuscript will end up being an interesting representation of how to develop new receptor-subtype selective and clinic-relevant ligands.

I suggest to improve a couple of issues in the manuscript:

- The authors could improve Figure 1c. I cannot see the salt bridge between haloperidol and D114 in this figure.

- This leads me to the next problem. The electron density for haloperidol is very poor (Figure S1a). I really don't find the chosen view sufficient either. I would suggest several things.

a) One should show two different views of haloperidol for the 2fo-fc.

b) Additionally one should show two views for haloperidol with a simulated-annealing omitted map.

c) Also, I can't distinguish the 2fo-fc map from the fo-fc map (best to separate them too).

d) The authors should include a figure in the SI to see all potential hydrogen-binding and hydrophobic interactions more precisely, e.g. a ligplot analysis for haloperidol. Comparing ligplots from the other ligands also for DRD3 and DRD4 would be very helpful.

- Which brings me straight to the next important point. The electron density looks like this, so if one could interpret the ligand haloperidol clearly also rotated by 180°. In order to clarify the orientation of the ligand, anomalous X-ray data could be collected. The Cl of the chlorobenzene should be well distinguishable from the F at higher wavelengths (> 1.8-2.0 Ang). Or are there other arguments for it?

- The concept of VBP and EBP does not seem very clear to me here. The risperidone and eticlopridie have completely different chemical groups in the potential EBP and a different moiety than the chlorobenzene from haloperidol. Of course this always means different interactions. I wouldn't describe this newly discovered binding site VBP as variable, there's no hint for that. If you take a closer look at figure 2e-h, the overall OBP's of DRD2-4 all seem to be more or less variable. The principle seems to be rather where you can influence the variability towards a certain receptor subtype. In other words the EBP for DRD2 is different. The authors could also calculate the surface interaction area for all ligands.

- Thus the sentence is also somewhat questionable and much too strongly formulated. There is no structure of DRD3 or 4 with haloperidol. "Although the Trp100EL1 of DRD3 and DRD4 locates at the same position as that in DRD2/haloperidol complex structure, the inward movement of EL2 in DRD3 and DRD4 forms a border of EBP in each receptor and then disrupts the formation of a similar VBP at DRD3 or DRD4 (Fig. 2c, d and Supplementary Fig. 3)." Another point is that if you enlarge the pocket in DRD3, you also get an enhancement of the haloperidol binding. If you reduce the size of the binding pocket in DRD4 by substituting L111F, you obtain increased Ki (less binding). This is simply said the same effect you see with F110A or F110L in DRD2 but on a different Ki -level. But, I agree with the authors that F/L 2.28 is a key element. Several other substitutions (e.g. W, Y, C, E) on this side chain would be very interesting to better understand the

key effect.

- What happens if you insert another bulky side chain into DRD2 like W100F?

- A sequence superposition of DRD2, DRD3 and DRD4 would be very helpful for the reader, especially for EL1 and EL2. How do these key elements differ in ligand binding (Sequence/lengths)? Figure S3 is simply not made understandable enough.

- The SBDD approach is relatively intuitive and attractive and one has found two interesting subtype-selective agonists in O4SE6 and O8LE6 but also ligands with a higher affinity for several subtypes. Why not try to make an X-ray structure of them? The information would really raise the description of the selective DRD2 receptor pocket to a better level with new information. The crystallization and production of DRD2 and the synthesis of the ligands does not seem to be a major issue. I would highly recommend that.

Response to Reviewers' Comments are in blue.

Reviewer #1 (Remarks to the Author):

In their manuscript entitled “Atypical structure-based discovery of selective D2 dopamine receptor ligands” by Luyu, Fan, Liang Tan, Zhangcheng Chen, Jianzhong Qi, Fen Nie, Zhipu Luo, Jianjun Cheng, and Sheng Wang, the authors address a topic that is very important to the field, the discovery and development of subtype-selective Dopamine D2 receptor (DRD2) ligands. One of the most important findings of the paper is a new X-ray crystal structure of the DRD2 in complex with the clinical antipsychotic haloperidol. This structure will be definitely appreciated by the community, as it has the potential to provide new insights into ligand binding at DRD2. However, it seems that this relevant result is over-shadowed by the attempt to develop novel DRD2 selective ligands employing a very unclear rationale. Even the title of the manuscript does not provide any hint to the new crystal-structure and this might lead to the publication being unnoticed by certain researchers in this field (like structural biologists, computer chemists etc.). Overall, the manuscript is not easy to read as it contains a lot of over-detailed description on data on one side, while on the other hand it is clearly lacking a good description of the rationale for the individual experiments performed. In its present form, the manuscript is thus not suitable for publication. A number of the logical and technical shortcomings are detailed below.

We thank the reviewer for all the in-depth comments, suggestions, and corrections.

To better highlight the new crystal structure, we have changed the title of the manuscript to “Haloperidol bound D₂ dopamine receptor structure inspired the discovery of subtype selective ligands”.

We also have reorganized the manuscript to enhance its readability. As can be also seen from the critiques from Reviewer 2, the manuscript was not easy to read due to the large number of compound descriptions. In our revised manuscript, we deleted all OLE analogs SAR data from main text and reorganized it in the Supplementary Note 1. We believe that revised manuscript reads better.

1) Unclear or rationale: The authors start with a brief description of their new crystal structure of DRD2 in complex with haloperidol which they refer to as DRD2-preferring. However, based on the binding data from Fig 1a it might not be justified. Furthermore, similar affinities to all D2-like receptors are demonstrated in IUPHAR/BPS Guide to Pharmacology. It is unclear what they intend to learn from this structure with a rather unselective ligand in the context of D2-selective ligands.

We referred to L-741626, not haloperidol, as a DRD2-preferring compound. Indeed, based on the binding data from Fig 1a, L-741626 is a DRD2-preferring compound, consistent with what has been reported by other groups¹⁻³. At the beginning, we tried to solve the DRD2/L-741626 complex structure. But we failed to get high-quality crystals. We then turned to haloperidol because its chemical structure is similar to L-741626. Furthermore, haloperidol is an important antipsychotic that has been used in the clinic for decades. We expected to obtain useful structural information of DRD2 selectivity from DRD2/haloperidol complex structure. Indeed, we have found a new extended binding pocket (previous as VBP) distinct from the risperidone-bound structure and a unique ligand binding pose. Based on this new structure, we have illuminated the key controllers for DRD2 selective activation.

Moreover, the differences to the previously published structure with risperidone are only sparsely discussed. How is it concluded, that DRD2 has a variable extended binding pocket (EBP) while those of DRD3 and DRD4 are considered to be rigid? Is this a new finding of the present study?

The big difference between the DRD2/haloperidol and DRD2/risperidone crystal structures is the second extended binding pocket (VBP in our original manuscript). To be more concise, we discussed only the structural differences around the ligand pocket between these two structures.

We agree with the reviewer that variable extended binding pocket (VBP) may be inappropriate. In the revised manuscript, we re-named it as the **second extended binding pocket (SEBP)**.

2) Can the authors explain why the extracellular sides of TM1 and TM2 in DRD2-haloperidol are shifted compared to the structure of DRD2-risperidone? Could it be just a crystallization artefact?

Based on the structures, we don't think it's a crystallization artefact. The observed difference of TM1 and TM2 is unlikely induced by crystal packing force (Supplementary Fig 2a, b). Because there are no observed direct contacts around the extracellular sides of TM1 and TM2 from neighboring molecules. We believe that the difference in the ligand binding poses (haloperidol vs risperidone) is the reason. Based on the structure (Fig A, as below), the residue Trp100^{EL1} in DRD2/haloperidol structure rotates outside of the binding pocket as compared to the risperidone-bound structure. The chlorobenzene moiety of the haloperidol apparently triggered the observed rotation of Trp100^{EL1} at D₂-like receptors. The rotation of Trp100^{EL1} shifted the EL1 and induced the different position of TM1 and TM2 between the haloperidol- and risperidone-bound DRD2 structures.

Fig. A. Comparison of the view from the extracellular side of a structural alignment DRD2/haloperidol (blue cartoon/orange stick) and DRD2/risperidone complex (green cartoon/yellow stick; PDB code: 6CM4).

3) Absence of actual structure-based design: instead of virtually screening the extended binding pocket of any of the DRD2 crystal structures for the discovery of selective ligands, the authors used ligands from a previous study with DRD4 that were characterized as high-ranking or mid-ranking molecules for DRD4. It remains unclear how this can be considered as a structure-inspired discovery of DRD2-selective ligands. The additional insight gained by obtaining the crystal structure of DRD2-haloperidol does not seem to be related to this part of the manuscript.

Based on DRD2/haloperidol complex structure, we identified the second extended binding pocket (SEBP), which forms by the junction of TM2 and TM3 and extends towards EL1. The previous DRD4 EBP reaches deep into a cleft between TM2 and TM3 of DRD4. Although these two pockets present critical differences, they locate in a similar position, but point to the different direction.

In our manuscript, we showed that residue 3.28 plays a key role in DRD2 versus DRD3/DRD4 selectivity. In our previous published paper⁴, we presented the 3.28 residue, which is leucine at DRD4, is also a key controller for DRD4 EBP.

Besides the similarity of the extended binding pockets, we also showed the similarity of the orthosteric binding pockets at DRD2 and DRD4. But, the orientations of ligands are completely different (Fig. 2e-h). The ligand in DRD2 is located deeper in the OBP and embedded in the deep binding pocket. In contrast, the ligands in DRD4 are located higher in the OBP and pointing to TM5, adopting a shallow binding mode (Fig. 2g, h).

“Thus, it would be a great advantage to identify DRD2- and/or DRD3- preferring compounds from these 164 high-ranking and 164 mid-ranking DRD4 unbound molecules, likely due to the critical difference of residue 3.28 between DRD4 (Leu111^{3,28}) and DRD2 (Phe110^{3,28}), the different orientation between DRD2 SEBP (points to an interface of TM3 and EL1) and DRD4 EBP (reaches deep into a cleft between TM2 and TM3) (Fig. 2a, d), and the different ligand binding poses at DRD2 versus DRD4 (deep versus shallow binding mode) (Fig. 2e, h).”

It is completely unclear how “this approach may have a broader application in accelerating the discovery of selective ligands” as claimed in the discussion.

Thank you for correcting this. We have now changed it as follows:

“Methodologically, the **combination of structure-based design and mechanism-driven screening** may have a broader application in accelerating the discovery of selective ligands to distinguish extremely similar receptors, which are still a large portion of drug targets.”

4) The authors claim to have “explored a previously unrecognized mechanism of DRD2 activation via the VBP” and suggest that the different binding pose (shallow vs deep) could be a key controller of DRD2 selectivity. From the present data, this conclusion does not appear to be fully justified, as the “shallow vs. deep” comparison between DRD2 and DRD3/4 includes ligands with very different molecular structure and size (benzamides nemonapride and eticlopride) vs. haloperidol/risperidone.

Although the ligands are different, the binding pose is determined by the shape of the pocket. Compared to the DRD3 and DRD4, the inward shift of TM5 in DRD2 shrinks the OBP substantially (Supplementary Fig. 5 and Fig. 2e-h). As a result, although the ligands bind to the same pocket-the OBP, their orientations are different, i.e. deep vs. shallow, between DRD2 and DRD3/DRD4 (Fig. 2e-h). Further, in the docking poses of haloperidol and risperidone at DRD3 or DRD4, the ligands adopted the shallow binding poses at DRD3 and DRD4 (**Fig. B**). And, the eticlopride and nemonapride adopted the deep binding pose at DRD2 (**Fig. B**). This is likely a direct consequence of the inward shift of TM5 at DRD2 (DRD2 vs DRD3/DRD4), respectively. (Supplementary Fig. 5). The data have been shown as **Supplementary Fig. 5e, f** in the manuscript. For clarity, we decided only show representative data of eticlopride and nemonapride docking poses at DRD2 in the manuscript for clarity reasons and to keep the focus on the following selective compound discovery.

Fig. B. Molecular docking of compounds into the D₂-like receptor structure. **a, d, h, p**, The crystallographic pose of compounds in the D₂-like receptor structure (**a**, DRD2/haloperidol; **d**, DRD2/risperidone; **h**, DRD3/eticlopride; **p**, DRD4/nemonapride). **b, c, e, f, g, i, j, o**, Docking of compounds in the D₂-like receptor structure. In all panels, receptors are shown as cartoon (DRD2 blue or green; DRD3 magenta; DRD4 tan). Ligands are shown as sticks (haloperidol orange; risperidone yellow; eticlopride cyan; nemonapride pink).

Moreover, these crystal structures were obtained with antagonists, and it is therefore hard to conclude whether the same differences would be obtained with agonists.

We understand this concern. The short answer is that we don't know the exact difference of ligand binding pockets between active and inactive states of D_2 -like receptors. Whereas GPCRs undergo striking conformational changes upon activation, in the area of the actual orthosteric site the changes are frequently modest, and rarely exceed 1 Å RMSD in main-chain atoms. For this reason, we think the difference in agonist binding poses at these receptors could be reflected by the antagonist-bound structures of DRD2 and DRD3/DRD4. To do it properly, one would need to solve the structures of agonist-bound, active states of D_2 -like receptors. We agree it would be great to do so, but it would be a major undertaking and would be beyond the scope of this paper.

5) The conclusions drawn from the SAR studies are peculiar, if not false. For example, the authors claim “For the 6 analogs (O1-6LE6) with various substitutions on the LHS benzofuran, the most potent analogs are O4 and O5 compounds, which have ethoxy and propoxy substituents respectively.” From the binding affinities shown in Supplementary table 4, it rather looks like there is no significant difference between O4, O5 and O6 and no correlation between the length of the alkyl chain and determined affinities. Also, the statement “For the RHS cyclohexane part (O4LE1-9), we found that the best substituents are cyclohexane (E6) and cycloheptane (E7)” is not accurate since the best results were obtained with O4LE8 and larger rings were not synthesized.

Indeed, there is no significant difference in binding affinities among O₄LE₆, O₅LE₆ and O₆LE₆ at DRD2. But, in the functional assays, O₄LE₆ (G $\alpha_{i/o}$ agonism EC₅₀=0.57 nM and β -arrestin2 recruitment EC₅₀=43.21 nM) and O₅LE₆ (G $\alpha_{i/o}$ agonism EC₅₀=0.51 nM and β -arrestin2 recruitment EC₅₀=47.96 nM) are ~10 times more potent than O₆LE₆ (G $\alpha_{i/o}$ agonism EC₅₀=46.42 nM and β -arrestin2 recruitment EC₅₀=557.3nM) (Supplementary Note Fig. 1b, c and Supplementary Table 5, 6).

It is also true for the RHS cyclohexane part (O₄LE₆₋₈). O₄LE₆ (G $\alpha_{i/o}$ agonism EC₅₀=0.57 nM and β -arrestin2 recruitment EC₅₀=43.21 nM) and O₄LE₇ (G $\alpha_{i/o}$ agonism EC₅₀=1.65 nM and β -arrestin2 recruitment EC₅₀=44.64 nM) are ~10 times more potent than O₄LE₈ (G $\alpha_{i/o}$ agonism EC₅₀=25.13 nM and β -arrestin2 recruitment EC₅₀=958.1 nM) (Supplementary Note Fig. 1d, e and Supplementary Table 5, 6).

We thank the reviewer for pointing out our inadequate description. We have now corrected that and included a better description.

“For the 6 analogs (**O1-6LE6**) with various substitutions on the LHS benzofuran, the most potent analogs are **O4** and **O5** compounds, which have ethoxy and propoxy substituents respectively (**based on the EC₅₀s of G $\alpha_{i/o}$ agonism and β -arrestin2 recruitment assays**) (Supplementary Note Fig. 1b, c, Supplementary Table 4-6 and Supplementary Note Table 1-3). For the RHS cyclohexane part (**O4LE1-9**), we found that the best substituents are cyclohexane (**E6**) and cycloheptane (**E7**) (**based on the EC₅₀s of G $\alpha_{i/o}$ agonism and β -arrestin2 recruitment assays**) (Supplementary Note Fig. 1b, c, Supplementary Table 4-6 and Supplementary Note Table 1-3).”

To enhance readability of the manuscript, we deleted all OLE analogs SAR data from main text and reorganized it in the Supplementary Note 1

6) It is astonishing that more than a 1000-fold difference in affinity (K_i = 1.9 μ M for O4LE6 to DRD2) and potency (EC₅₀ = 0.57 nM) is obtained. This should be addressed thoroughly in the results section of the manuscript.

The use of the antagonist radioligand [³H]-N-methylspiperone could explain the difference between binding K_i and functional EC₅₀ values. The binding affinity of an agonist to an uncoupled GPCR normally appears as very low, because the uncoupled GPCR is in an inactive state that has traditionally been referred to as the ‘low-affinity’ state⁵⁻⁸. This ‘low-affinity’ state was also observed with the control compound dopamine in these assays, which showed a K_i of 1.64 μ M and an EC₅₀ of 0.20 nM (G $\alpha_{i/o}$ -mediated cAMP inhibition assay), respectively (Supplementary Table 3).

In the revised manuscript, we have one paragraph addressing this point.

“Since the EC₅₀ value in the G $\alpha_{i/o}$ -mediated cAMP inhibition assay may be influenced by the signal amplification, **O4LE6** was subjected to an orthologous nonamplification assay of DRD2 G protein activity measuring G $\alpha_{i/o}$ - γ 2 dissociation by bioluminescent resonance energy transfer (BRET). In this assay, **O4LE6** showed modest agonist activity (EC₅₀ = 24.14 nM, Supplementary Table 3), recapitulating our findings obtained from measuring G $\alpha_{i/o}$ -mediated cAMP inhibition activity. The use of the antagonist radioligand [³H]-N-methylspiperone could explain the difference between binding K_i and functional EC₅₀ values, which is consistent with previous results demonstrating that the affinity of agonists for an uncoupled GPCR (traditionally been referred to as the ‘low-affinity’ state) would appear very low^{10,20-22}. This ‘low-affinity’ state was also observed with the control compound dopamine in these assays, which showed a K_i of 1.58 μ M but an EC₅₀ of 0.20 nM (G $\alpha_{i/o}$ -mediated cAMP inhibition assay) or 50.37 nM (BRET), respectively (Supplementary Table 3).”

7) The authors claim that their work yielded highly selective DRD2 agonists, compounds O4SE6 and O8LE6. This key finding is only supported by functional assays and surprisingly ligand binding assays to determine the affinities of these compounds were not performed. Moreover, selectivity was only investigated in the TANGO- β -arrestin assay using agonist mode, neglecting the fact that these compounds could act as antagonists of other receptors.

We have now included the affinities of these compounds as **Supplementary Table 4** (using the antagonist radioligand [3 H]-N-methylspiperone). Just as we have mentioned above, by using of the antagonist radioligand [3 H]-N-methylspiperone we could not get the accurate agonists' affinities at DRD2. So, it is not surprise that the 'affinities' of compounds O₄SE₆ and O₈LE₆ didn't show any selectivity towards DRD2. However, data from the functional assays show O₄SE₆ and O₈LE₆ are definitely DRD2 highly selective agonists. It would be impracticable to check antagonist activity of O₄SE₆ or O₈LE₆ for each receptor. In our revised manuscript, we screened all dopamine receptors and serotonin receptors using both agonist and antagonist modes (as dopamine ligands often have off target effects at serotonin receptors), and have now included a new **Supplementary Fig. 10**. Based on these results, O₄SE₆ and O₈LE₆ didn't show antagonist activity at dopamine receptors and serotonin receptors. Beside agonist activity at DRD2, O₄SE₆ and O₈LE₆ activate 5HT_{1D} and 5HT_{7A} receptors only at high concentrations (over 1 μ M).

8) Misleading abstract and introduction: Form the abstract and the introduction the reader gets the impression that non subtype-selective dopamine receptor antagonists cause life-threatening side effects by interaction with DRD3 or DRD4, while the references indicated actually refer e.g. to valvular heart disease caused by 5-HT_{2B} receptors.

We have now corrected the abstract.

“The D₂ dopamine receptor (DRD2) is one of the most well-established therapeutic targets for neuropsychiatric and endocrine disorders. **Most clinically approved and investigational drugs that target this receptor are known to be subfamily-selective for all three D₂-like receptors rather than subtype-selective for only DRD2.** Here we reported the crystal structure of DRD2 bound to the most commonly used antipsychotic drug-haloperidol. The structures suggest a previously unrecognized extended binding pocket for DRD2 that distinguish it from other D₂-like subtypes. A detailed analysis of the structures illuminated key structural determinants essential for DRD2 activation and subtype selectivity. A structure-based and mechanism-driven screening combined with a lead optimization approach yielded DRD2 highly selective agonists, which could be used as chemical probes for studying the physiological and pathological functions of DRD2 as well as promising therapeutic leads devoid of promiscuity.”

We have also corrected the misleading introduction.

“Most DRD2 ligands concomitantly bind to the DRD3 or/and DRD4. Thus, there is a desire to develop compounds that selectively target the DRD2 with minimal subtype cross-reactivity, and to ascertain the physiological and pathological functions governed by DRD2.”

9) Wrong referencing to Figures and Tables throughout the text: on various occasions, the authors describe results and reference to Figures that actually do not contain this data. E.g., “The docking pose of O4SE6 and O4RE6 at DRD2 showed that these ligands also adopted the same binding pose as haloperidol or risperidone in the complex (Figure S8a/b).” Instead, Figure S8 compares DRD2-haloperidol and DRD3-eticlopride.

Thank you, we have now added the right reference.

“The docking pose of **O₄SE₆** and **O₄RE₆** at DRD2 showed that these ligands also adopted the same binding pose as haloperidol or risperidone in the complex (Supplementary Fig. 7a, b and **Fig. 2e, f**). “

10) Use of tables: the numerous different tables containing very similar results are confusing to the reader e.g. Tables S4 and S7 both contain cAMP inhibition signaling and could be combined. Sometimes, the same data is presented in one table even more than once, e.g. O4LE6 in Table S4.

As suggested by the reviewer, we combined the similar results into one table and reorder the tables. And, we also deleted the repeated results.

11) Technical issues

- The chemical procedures lack general information on the purchase of reagents, instruments used etc.
- No information on test compound purity is provided. This is critical when discussing SAR.
- The experimental methods describing the synthesis of test compounds are lacking sufficient details allowing repetition of the procedure. For each compound at least the exact amount of reagents (mmols, equivalents) and solvents as well as the yields should be given.

We apologize for the lack of the necessary information. We have now corrected that and included a better description of the methods.

- The nomenclature of the compounds O4LE6 vs O5LE6 etc. is confusing, as the reader needs to pay very detailed attention to the individual numbers

We think the naming system may be easier for reader to understand. O means OBP binding moiety. E means EBP binding moiety. For clarity, we decided only show O₄LE₅₋₈ and O₇₋₈LE₆ in the main text of revised manuscript and move the rest analogs information to Supplementary Note 1.

- The description of the biological data is partly lacking important details, e.g. how were the cells transfected (reagent, cDNA vectors etc.)

We apologize for the lack of describing the transfection method sufficiently. We have now corrected that and included a better description of the method.

- The composition of all the assay buffers and the origin of commercially available reagents should be specified. For example, where does the His-tagged TEV protease come from?

We apologize for the lack of the necessary information. We have now specified the origin of commercially available reagents.

- Comparison of the expression levels of mutated receptors: the authors conclude similar expression levels from receptor constructs that were largely modified on their C-termini (additional expression of RLuc). Moreover, the authors directly measure RLuc activity after addition of the Coelenterazine substrate. Usually, an incubation time is needed to assure that Coelenterazine is able to cross the cell membrane, as the receptor C-terminus is and thus the luciferase is located intracellularly. The authors should use saturation binding to compare the expression of the individual receptors.

We thank the reviewer for pointing out the errors in the description of methods. As suggested by the reviewer, we used the saturation binding to compare the expression level of different DRD2 mutants (**Fig. C**). Compared to the wide-type receptor, only F110A^{3,28} mutant shows slightly reduced expression level (**Fig. C**). The results are now summarized in **Supplementary Fig. 7c** in the manuscript, and the raw data shown in **Fig. C** below due to limited space in the manuscript.

Fig. C. Measurement of saturation binding at different DRD2 mutants with ³H-methylspiperone. Data represent means ± s.e.m. from three independent experiments.

- While presenting binding curves or functional assays it should be clearly stated whether the displayed graph is obtained from a pooled data or is shown as a representative graph from a single experiment (Fig. 1a etc.).

The data is obtained from three independent experiments. We now fixed our mistakes (from 'N=3' to 'n=3 independent experiments').

- Representation of the amino acid mutations should be carefully thought through. At the moment many different variants are used for one position (like F3.28A, F1103.28A, Phe1103.28Ala.). This seems to be also confusing the authors, since sometimes the mutations are referred to in the text incorrectly.

We thank the reviewer for pointing out our mistakes. We have made Phe110^{3.28} for the main text and F110^{3.28} for the figures.

- The authors are encouraged to think about how many significant figures are appropriate for measurement results. For consistency of reporting, the same number of significant figures should be used within an assay (at the moment it varies from 1 to 6).

We would like to thank the reviewer's suggestion. We unified the significant figures in our revised manuscript.

- There are numerous grammatical and spelling errors throughout the text.

We have now improved the flow of the manuscript and corrected some grammatical and spelling errors.

Reviewer #2 (Remarks to the Author):

The manuscript "Atypical structure-based discovery of selective D2 dopamine receptor ligands" by Sheng Wang and colleagues has very interesting results and could potentially promote new development of dopamine receptor drugs. The authors investigated the D2 dopamine receptor (DRD2), which is one of the most important therapeutic targets in neuropsychiatric diseases. No real DRD2-selective drugs are available yet. The authors showed a new (so-called variable part or VBP) extension of the orthosteric binding pocket at DRD2, which differs from other D2-like subtypes (DRD3 or DRD4) and can be used for the optimization of new ligands, based on the crystal structure of a DRD2-haloperidol complex. Using a structure-based design approach, they found highly selective agonists for DRD2, which could play an important role for highly specific binding and downstream signaling in the future.

The manuscript unfortunately reads very technically in many parts and is very descriptive, which is certainly due to the large number of compound descriptions.

The methods (assays and X-ray crystallography) used for investigation are state-of-the-art and of high quality.

I cannot evaluate the chemical synthesis of these compounds.

I believe the manuscript will end up being an interesting representation of how to develop new receptor-subtype selective and clinic-relevant ligands.

We thank the reviewer's appreciation. To enhance readability of the manuscript, we deleted all OLE analogs SAR data from main text and reorganized it in the Supplementary Note 1.

I suggest to improve a couple of issues in the manuscript:

- The authors could improve Figure 1c. I cannot see the salt bridge between haloperidol and D114 in this figure.

As suggested by the reviewer, we added a new figure (**Supplementary Fig. 1f**) to highlight the salt bridge.

- This leads me to the next problem. The electron density for haloperidol is very poor (Figure S1a). I really don't find the chosen view sufficient either. I would suggest several things.

a) One should show two different views of haloperidol for the 2fo-fc.

b) Additionally one should show two views for haloperidol with a simulated-annealing omitted map.

c) Also, I can't distinguish the 2fo-fc map from the fo-fc map (best to separate them too).

d) The authors should include a figure in the SI to see all potential hydrogen-binding and hydrophobic interactions more precisely, e.g. a ligplot analysis for haloperidol. Comparing ligplots from the other ligands also for DRD3 and DRD4 would be very helpful.

We thank the reviewer for pointing out our inadequate coverage of the electron density, and have now included a new **supplementary Fig. 1** with a much more comprehensive display of the electron density and added ligplot analysis as **supplementary Fig. 3f**. We didn't include the ligplots of DRD3 and DRD4. The ligplots already published in the previous paper^{4,9}.

- Which brings me straight to the next important point. The electron density looks like this, so if one could interpret the ligand haloperidol clearly also rotated by 180°. In order to clarify the orientation of the ligand, anomalous X-ray data could be collected. The Cl of the chlorobenzene should be well distinguishable from the F at higher wavelengths (> 1.8-2.0 Å). Or are there other arguments for it?

Based on the electron density, it's impossible to rotate haloperidol by 180°. If the haloperidol rotated by 180°, the error density map (red) indicated the ligand is in the wrong pose (**Fig. D**). And, the previous position of piperidine ring shows green density (**Fig. D**). It means there is some atoms need to build. By the way, based on previous published DRD2 structure, the F of risperidone also adopted the same position as F of haloperidol.

Fig. D. The electron density map for the mismatch haloperidol pose at DRD2. The 2Fo-Fc electron density map is colored blue and contoured at 1.0 σ . The Fo-Fc omit map is colored green (3.0 σ) and red (-3.0 σ); the green mesh shows where atoms are missing in the current model, while the red mesh shows where atoms are present in the model but not the crystal.

- The concept of VBP and EBP does not seem very clear to me here. The risperidone and eticloprodie have completely different chemical groups in the potential EBP and a different moiety than the chlorobenzene from haloperidol. Of course this always means different interactions. I wouldn't describe this newly discovered binding site VBP as variable, there's no hint for that. If you take a closer look at figure 2e-h, the overall OBP's of DRD2-4 all seem to be more or less variable. The principle seems to be rather where you can influence the variability towards a certain receptor subtype. In other words the EBP for DRD2 is different. The authors could also calculate the surface interaction area for all ligands.

We agree with the reviewer that variable extended binding pocket (VBP) may be not appropriate. In the revised manuscript, we named it as **second extended binding pocket (SEBP)**.

- Thus the sentence is also somewhat questionable and much too strongly formulated. There is no structure of DRD3 or 4 with haloperidol. "Although the Trp100EL1 of DRD3 and DRD4 locates at the same position as that in DRD2/haloperidol complex structure, the inward movement of EL2 in DRD3 and DRD4 forms a border of EBP in each receptor and then disrupts the formation of a similar VBP at DRD3 or DRD4 (Fig. 2c, d and Supplementary Fig. 3)."

Thank you. We changed the sentence as follow.

"Although the conserved residue Trp^{EL1} of DRD3 and DRD4 locates at the same position as that in DRD2/haloperidol complex structure, the inward movement of EL2 in DRD3 and DRD4 forms a border of EBP in each receptor (Fig. 2c, d and Supplementary Fig. 3a-e). And, the different position of EL2 at D₂-like receptors is unlikely induced by the crystal packing forces (Supplementary Fig. 2c, d). Based on the comparison, DRD2 SEBP partially overlap with the previous identified DRD3 EBP, which consists of TM2, TM7, EL1 and EL2 (Fig. 2a, c). Compared to the DRD3, an outward movement of EL2 makes additional space for DRD2 SEBP (Fig. 2c, d and Supplementary Fig. 3c)."

Another point is that if you enlarge the pocket in DRD3, you also get an enhancement of the haloperidol binding. If you reduce the size of the binding pocket in DRD4 by substituting L111F, you obtain increased K_i (less binding). This is simply said the same effect you see with F110A or F110L in DRD2 but on a different K_i -level. But, I agree with the authors that F/L 2.28 is a key element. Several other substitutions (e.g. W, Y, C, E) on this side chain would be very interesting to better understand the key effect.

We also followed the reviewer's advice and measured the other substitutions of 3.28 (**Table A**). To summary, if you increase the size of 3.28 side chain (e.g. W, Y), the affinity of haloperidol and L-741626 decreased. If you decrease the size (e.g. A), the affinity of haloperidol and haloperidol increased. We decided not to include this data in the manuscript for clarity reasons and to keep the focus on F110^{3.28A}.

Table A | Affinity of L-741,626 and haloperidol at mutants of the D₂ dopamine receptor.

Receptor	L-741,626		Haloperidol	
	K _i , nM (pK _i ± SEM)	ΔpK _i (mutant-WT)	K _i , nM (pK _i ± SEM)	ΔpK _i (mutant-WT)
DRD2 wild-type	15.85 (7.80±0.15)	--	0.46 (9.34±0.09)	--
DRD2 F110 ^{3,28} W	134.89 (6.87±0.39)	-0.93	12.88 (7.89±0.14)	-1.45
DRD2 F110 ^{3,28} Y	912.01 (6.04±0.44)	-1.76	19.05 (7.72±0.05)	-1.62
DRD2 F110 ^{3,28} C	3.89 (8.41±0.47)	0.61	0.45 (9.35±0.21)	0.01
DRD2 F110 ^{3,28} E	14.79 (7.83±0.10)	0.03	0.79 (9.10±0.35)	-0.24
DRD2 F110 ^{3,28} A	0.11 (9.94±0.02)	2.14	0.03 (10.52±0.06)	1.18
DRD2 W100 ^{ECL1} F	112.20 (6.95±0.33)	-0.85	13.18 (7.88±0.17)	-1.46
DRD2 W100 ^{EL1} A	97.72 (7.01±0.26)	-0.79	1.51 (8.82±0.17)	-0.52

Data represent mean K_i (pK_i ± SEM) for competition binding experiments using [³H]-methylspiperone(0.3-0.5 nM) as radioligand. All data are the mean ± SEM of three independent assays (n = 3 independent experiments).

- What happens if you insert another bulky side chain into DRD2 like W100F?

The binding affinities of haloperidol and L-741626 to W100^{EL1}F mutant of DRD2 also decreased (Table A).

- A sequence superposition of DRD2, DRD3 and DRD4 would be very helpful for the reader, especially for EL1 and EL2. How do these key elements differ in ligand binding (Sequence/lengths)? Figure S3 is simply not made understandable enough.

As suggested by the reviewer, we changed the Supplementary Fig. 3 and highlighted the key elements.

- The SBDD approach is relatively intuitive and attractive and one has found two interesting subtype-selective agonists in O4SE6 and O8LE6 but also ligands with a higher affinity for several subtypes. Why not try to make an X-ray structure of them? The information would really raise the description of the selective DRD2 receptor pocket to a better level with new information. The crystallization and production of DRD2 and the synthesis of the ligands does not seem to be a major issue. I would highly recommend that.

We agree it would be great to solve the new structures, but it would be a major undertaking and would be beyond the scope of this paper. We thank the reviewer pointed out this important future direction.

References

- 1 Vangveravong, S. *et al.* Synthesis and characterization of selective dopamine D2 receptor antagonists. *Bioorg Med Chem* **14**, 815-825, (2006).
- 2 Grundt, P., Husband, S. L., Luedtke, R. R., Taylor, M. & Newman, A. H. Analogues of the dopamine D2 receptor antagonist L741,626: Binding, function, and SAR. *Bioorg Med Chem Lett* **17**, 745-749, (2007).
- 3 Xiao, J. *et al.* Discovery, optimization, and characterization of novel D2 dopamine receptor selective antagonists. *J Med Chem* **57**, 3450-3463 (2014).
- 4 Wang, S. *et al.* D4 dopamine receptor high-resolution structures enable the discovery of selective agonists. *Science* **358**, 381-386 (2017).
- 5 Lahti, R. A. *et al.* Dopamine D2 receptor binding properties of [3H]U-86170, a dopamine receptor agonist. *Eur J Pharmacol* **202**, 289-291 (1991).
- 6 Zou, M. F. *et al.* Novel Analogues of (R)-5-(Methylamino)-5,6-dihydro-4H-imidazo[4,5,1-ij]quinolin-2(1H)-one (Sumanitrole) Provide Clues to Dopamine D2/D3 Receptor Agonist Selectivity. *J Med Chem* **59**, 2973-2988 (2016).
- 7 Hino, T. *et al.* G-protein-coupled receptor inactivation by an allosteric inverse-agonist antibody. *Nature* **482**, 237-240 (2012).
- 8 Staus, D. P. *et al.* Allosteric nanobodies reveal the dynamic range and diverse mechanisms of G-protein-coupled receptor activation. *Nature* **535**, 448-452 (2016).
- 9 Chien, E. Y. *et al.* Structure of the human dopamine D3 receptor in complex with a D2/D3 selective antagonist. *Science* **330**, 1091-1095 (2010).

Reviewers' Comments:

Reviewer #1:

Remarks to the Author:

The revised version of the manuscript "Haloperidol-bound D2 dopamine receptor structure inspired the discovery of subtype-selective ligands" by Sheng Wang and colleagues describes the discovery of subtype-selective DRD2 ligands inspired by a new DRD2 crystal structure with the clinically used antipsychotic haloperidol as co-crystallized ligand.

Compared to the initial submission, the manuscript has been largely improved, especially concerning its readability and the emphasis that is now attributed to the novel crystal structure. In their revised version, the authors have addressed most of my comments and - provided that the remaining minor issues detailed below have been properly addressed - the manuscript is recommended for publication.

Introduction

Line 67: "Significantly, we found that the SEBP not only directly interacts with the DRD2-preferring compound, but also plays a key role in DRD2 agonist activation."

At this point, it is unclear to the reader, what "the DRD2 preferring compound" is, since L-741626 has not been introduced. In addition, from the context, one could get the impression, that the authors refer to an L-741626 crystal structure, which is not the case. Please modify accordingly.

Results

Lines 136-138: "[...] greater affinity enhancement at Phe106 (3.28) Ala mutant of DRD2"

The correct numbering of the amino acid within the DRD2 in this position is Phe110. Moreover, the exact same information has already been given in lines 123-125.

Lines 142-143: While this manuscript was under revision, a DRD4 crystal structure in complex with the DRD4 selective ligand L-745870 (with a highly similar chemical structure) was published (Zhou et al. eLife 2019;8: e48822). The authors are encouraged to include this information in their manuscript.

Lines 190-193: The authors now appropriately discuss the divergence between binding affinity and functional potency of O4LE6. However, they should also mention O4LE6's affinity in the text, not only the range of affinities of all their hits (line 180).

Lines 215-216: While the authors give information on the expression level of the mutant, they could also add information on the behavior of the reference agonist (how does its EC50 change?) to elucidate whether Phe3.28 is critical for DRD2 activation by all agonists, or OLE compounds only.

Line 227: "Molecular docking of O4SE6 and O4RE6 [...]"

It would be beneficial for an easy understanding to mention that O4LE6 is the racemic mixture of O4SE6 and O4RE6 and that the individual enantiomers were docked in order to understand their mechanism of action.

Lines 256-260: "Unexpectedly, when we compared the G-protein agonist activity of O4SE6 and O4RE6, we observed that O4SE6 retained G-protein signaling agonist activity at DRD2 and lost its efficacy at DRD3 [...]"

The way the authors describe the functional activity of the two enantiomers in this section, it is not clear what comparisons they actually want to make. What do they mean by "retained/lost its activity"? Based on the data, they try to compare the individual behavior of the enantiomers with the racemic mixture at the DRD2 and DRD3 wildtype receptors. I cannot see how O4RE6 "dramatically lost its efficacy at DRD3" since it shows an Emax of 52% which is close to the effect of the racemic mixture O4LE6 (62%). This is especially confusing, as it contradicts the postulation made in lines 233-235, stating both ligands have similar behaviors at DRD2 and its Phe110 (3.28)

mutants. I would also suggest moving the passage 256-260 into lines 233 and following, as it is unclear, why the authors return to the wildtype receptors after having brought up all the mutational investigations.

Line 268: Again, use of the expression "lost activity". A compound cannot lose an activity it never had.

Discussion

Lines 322-324: "Because [...]" This is not a complete sentence.

Figures

Figure 4: "optically pure enantiomers" is an unusual expression. The authors could use "individual enantiomers" or "enantiopure compounds" instead.

Figure 5: The title appears misleading to me, since the figure shows mostly assay data and no structure-based or mechanism-driven design.

Supplementary Figure S7: The figure shows docking of ORE and OSE compounds and not OLE compounds. Please correct the title.

Reviewer #2:

Remarks to the Author:

The authors provided satisfactory explanations in part to my comments.

Several additional comments:

- I cannot find a ligplot analyses for haloperidol to see all potential hydrogen-binding and hydrophobic interactions in a -real- plot (SI3f is not satisfactory. There are no differences between van der Waals contacts or H-bonds).
- I suggest using a simulated-annealing omitted map at 3.0 sigma level, which is a default level for a fo-fc map type to control the quality of density.
- Have the authors even used a simulated-annealing omitted map? They call it fo-fc omit map which could be the wrong one. With the simulated-annealing omitted map the reader could see an almost unbiased map, which is essential for the contact analyses.
- The authors should make it clearer that there is no direct contact for the Cl of chlorobenzene moiety in haloperidol. This is clearly visible in the omitted map due to lack of density.
- Regarding to the comment of Reviewer 1: I cannot see any crystal contacts in Supplementary Figure 2. "Crystal packing of D2-like receptors". Unfortunately, this figure does not give this statement. Is there any crystal contact (hydrophobic or H-bonds) in the whole ECL1 or at the end of TM2, TM1 or TM3 to the symmetry-related T4L or other parts of the protein construct? If so, the authors should clarify this in the main text. That would be fair to the reader. I also clearly see the differences in ECL1 between DRD2/haloperidol and DRD2/risperidone complexes, but a crystal contact can cause a specific local state (which does not necessarily mean that it has to be artificial). The same is true for the situation with DRD2/risperidone. Is there a crystal contact in the above region? In the manuscript is only a statement like "is unlikely induced by the crystal packing forces".
- I suggest to include the mutant data for the variants DRD2 F110W and F110Y in the manuscript. These data underline the fact that the size of the binding pocket, which is reduced by amino acids larger than F, is crucial, reflected in a decreased Ki for L-741.626 and haloperidol. It becomes really interesting when you compare DRD2 F1103.28C and F1103.28E. Both amino acids have rather similar sizes but completely different properties. Both mutants have almost similar Ki as the WT for both ligands. Here only the size of the amino acid seems to play a role for the binding affinity. This statement is finally strengthened with the variant DRD2 F1103.28A (increased Ki compared to the WT). These data simply belong at least in the supplement.
- The DRD2 W100ECL1F variant is also very interesting. It gives a hint that L-741.626 and

haloperidol are in a slightly different conformation in the binding pocket (shallow vs deep?). The 28-fold decreased K_i in DRD2 W100ECL1F with haloperidol (compared to the WT) is an effect, which is partly seen with L-741.626 (8-fold). However, when the mutant DRD2 W100ECL1A is used, both ligands behave differently. With L-741.626 the K_i is 6-fold lowered similar to the DRD2 W100ECL1F variant, but with haloperidol only 3-fold. These data should also be mentioned.

Response to Reviewers' Comments are in blue.

Reviewer #1 (Remarks to the Author):

The revised version of the manuscript "Haloperidol-bound D2 dopamine receptor structure inspired the discovery of subtype-selective ligands" by Sheng Wang and colleagues describes the discovery of subtype-selective DRD2 ligands inspired by a new DRD2 crystal structure with the clinically used antipsychotic haloperidol as co-crystallized ligand.

Compared to the initial submission, the manuscript has been largely improved, especially concerning its readability and the emphasis that is now attributed to the novel crystal structure. In their revised version, the authors have addressed most of my comments and - provided that the remaining minor issues detailed below have been properly addressed - the manuscript is recommended for publication.

We thank the reviewer for these positive comments. We have made changes to address all remaining minor issues and we believe that revised manuscript is now suitable for publication.

Introduction

Line 67: "Significantly, we found that the SEBP not only directly interacts with the DRD2-preferring compound, but also plays a key role in DRD2 agonist activation."

At this point, it is unclear to the reader, what "the DRD2 preferring compound" is, since L-741626 has not been introduced. In addition, from the context, one could get the impression, that the authors refer to an L-741626 crystal structure, which is not the case. Please modify accordingly.

Thank you for correcting this. We have now changed it as follows:

"To address this problem, we solved the complex structure of the DRD2 bound to a commonly used typical antipsychotic drug-haloperidol. **Haloperidol is a potent antagonist of the DRD2 and it shares a substructure with the reported DRD2-preferring compound L-741626 (Fig. 1a).** Analysis of the DRD2-haloperidol complex structure revealed an unexpected second extended binding pocket (SEBP). Significantly, we found that the SEBP not only **directly interacts with haloperidol**, but also plays a key role in DRD2 agonist activation"

Results

Lines 136-138: "[...] greater affinity enhancement at Phe106 (3.28) Ala mutant of DRD2"

The correct numbering of the amino acid within the DRD2 in this position is Phe110. Moreover, the exact same information has already been given in lines 123-125.

We thank the reviewer for pointing this out. This has been corrected.

Lines 142-143: While this manuscript was under revision, a DRD4 crystal structure in complex with the DRD4 selective ligand L-745870 (with a highly similar chemical structure) was published (Zhou et al. eLife 2019;8: e48822). The authors are encouraged to include this information in their manuscript.

We would like to thank the reviewer's suggestion. We have now added the information of DRD4/L-745870 complex structure in our revised manuscript.

"The ligand in DRD2 is located deeper in the OBP and embedded in the deep binding pocket defined by the side chains of TM3, TM5 and TM6, which accommodates the butyrophenone moiety of haloperidol (Fig. 2e) and benzisoxazole moiety of risperidone (Fig. 2f). By contrast, the ligands in DRD3 and DRD4 are located higher in the OBP, pointing to TM5, adopting a shallow binding mode (Fig. 2g, h). **In the recent published paper, the L-745870 which share a similar chemical structure with haloperidol and L-741626, also adopts a similar shallow binding mode, not the deep binding mode, in the DRD4/L-745870 complex structure.**"

Lines 190-193: The authors now appropriately discuss the divergence between binding affinity and functional potency of O4LE6. However, they should also mention O4LE6's affinity in the text, not only the range of affinities of all their hits (line 180).

We thank the reviewer for pointing out our inadequate description. We have now corrected that and included a more detailed information of O₄LE₆'s affinity.

“Indeed, four compounds were identified from these 328 compounds, which showed DRD2 and/or DRD3 binding affinities (K_i) ranging from 0.21 μM to 4.27 μM in a competition binding assay with the antagonist radioligand [³H]-N-methylspiperone (Supplementary Table 3 and Supplementary Fig. 4). **Notably, compound 540595123 (O₄LE₆) showed the binding affinity for DRD2 and DRD3 (K_i=1.91 μM for DRD2 / 0.22 μM for DRD3, Supplementary Table 3), and also displayed potent agonist activity in a following DRD2 G_{α_{i/o}}-mediated cAMP inhibition assay (EC₅₀ = 0.57 nM, Fig. 3a and Supplementary Table 3).”**

Lines 215-216: While the authors give information on the expression level of the mutant, they could also add information on the behavior of the reference agonist (how does its EC₅₀ change?) to elucidate whether Phe^{3.28} is critical for DRD2 activation by all agonists, or OLE compounds only.

We have now added the data of the reference compound quinpirole in our revised manuscript. It could be observed that the activity of the OLE compounds (which interact with Phe^{110^{3.28}}) was greatly influenced by the mutations, but the agonist activity of quinpirole was largely maintained, with only slight changes of potency.

“With the Phe^{110^{3.28}}Ala mutant, O₄LE₆₋₈ failed to activate G_{α_{i/o}} or to recruit β-arrestin2 (Fig. 3 and Supplementary Table 5, 6), indicating that the Phe^{110^{3.28}}Ala substitution disrupts both G_{α_{i/o}} and β-arrestin2 agonism. Whereas the Phe^{110^{3.28}}Leu substitution led to O₄LE₆₋₈'s partial activation of G_{α_{i/o}} and β-arrestin2 signaling pathways with respect to full agonist quinpirole (Fig. 3, Supplementary Fig. 6a, b and Supplementary Table 5, 6). **In the Phe^{110^{3.28}}Ala/Leu mutants, quinpirole showed similar agonist activity in both G_{α_{i/o}} and β-arrestin2 signaling pathways as that in the wild type although with slightly reduced potency (EC₅₀=0.20 nM, 3.39 nM and 4.04 nM for wild type, Phe^{110^{3.28}}Ala and Phe^{110^{3.28}}Leu in G_{α_{i/o}} signaling, and EC₅₀=7.34 nM, 210.20 nM and 44.17 nM in β-arrestin2 recruitment respectively) (Supplementary Table 5, 6).”**

Line 227: “Molecular docking of O₄SE₆ and O₄RE₆ [...]”

It would be beneficial for an easy understanding to mention that O₄LE₆ is the racemic mixture of O₄SE₆ and O₄RE₆ and that the individual enantiomers were docked in order understand their mechanism of action.

We thank the reviewer for this suggestion. We have corrected this and included a more detailed description of **O₄LE₆, O₄SE₆ and O₄RE₆**.

“**O₄LE₆ is a racemic mixture of O₄SE₆ and O₄RE₆ and that the individual enantiomers were docked in order to understand their mechanism of action.** Molecular docking of **O₄SE₆ and O₄RE₆** to the DRD2/haloperidol and DRD2/risperidone crystal structures showed that only DRD2/haloperidol crystal structure could recapture the interaction between Phe^{110^{3.28}} and RHS cyclohexane moiety (Supplementary Fig. 7a, b).”

Lines 256-260: “Unexpectedly, when we compared the G-protein agonist activity of O₄SE₆ and O₄RE₆, we observed that O₄SE₆ retained G-protein signaling agonist activity at DRD2 and lost its efficacy at DRD3 [...]”

The way the authors describe the functional activity of the two enantiomers in this section, it is not clear what comparisons they actually want to make. What do they mean by “retained/lost its activity”? Based on the data, they try to compare the individual behavior of the enantiomers with the racemic mixture at the DRD2 and DRD3 wildtype receptors. I cannot see how O₄RE₆ “dramatically lost its efficacy at DRD3” since it shows an E_{max} of 52% which is close to the effect of the racemic mixture O₄LE₆ (62%).

We apologize for the unclear statements. Actually, we want to compare the functional activity of the individual enantiomer O₄SE₆ and O₄RE₆ with the racemic mixture O₄LE₆. O₄LE₆ showed agonism activity at both DRD2 and DRD3. And, the individual enantiomer O₄SE₆ showed agonism activity only at DRD2, not at DRD3. However, we would like to modify the description in a way that is easier to understand in the

revised manuscript. In addition, we agree with the reviewer that the word dramatically may be not appropriate. We have replaced the words “dramatically lost efficacy” with the words “**a lower efficacy**”.

“Unexpectedly, when we compared the G-protein agonist activity of **O₄SE₆** and **O₄RE₆** with **O₄LE₆** at DRD2 and DRD3, we observed that **O₄SE₆ showed G-protein signaling agonist activity** at DRD2 ($G\alpha_{i/o}$ agonism $EC_{50}=18.45$ nM) and **no detectable agonist activity** at DRD3; and, **O₄RE₆** displayed agonist activity at both receptors and had **a lower efficacy** at DRD3 ($G\alpha_{i/o}$ signaling $E_{max}=94.46\%$ for DRD2 / 52.44% for DRD3) (Fig. 4d, f and Supplementary Table 5). “

This is especially confusing, as it contradicts the postulation made in lines 233-235, stating both ligands have similar behaviors at DRD2 and its Phe110 (3.28) mutants. I would also suggest moving the passage 256-260 into lines 233 and following, as it is unclear, why the authors return to the wildtype receptors after having brought up all the mutational investigations.

In the lines 233-235, we discussed the residue Phe110^{3.28} is one of the key elements for the O₄LE₆ agonism activity at DRD2. However, in the lines 256-260, we discussed other important elements (the conserve serine residues) for the O₄LE₆ agonism activity at DRD2. We don't think it suitable for moving the passage 250-260 into lines 233. Besides, the lines 233 and above focus on the activation mechanism through SEBP. And, the line 256-260 and following focus on the OBP activation mechanism.

Line 268: Again, use of the expression “lost activity”. A compound cannot lose an activity it never had.

This have been corrected accordingly:

“**O₄SE₆** showed agonist activity at DRD2 (β -arrestin2 recruitment $EC_{50}=1055$ nM) and **no detectable agonist activity** at DRD3; and, **O₄RE₆** displayed different efficacy agonist activity at both receptors (β -arrestin2 recruitment $E_{max}=99.61\%$ for DRD2 / 21.41% for DRD3) (Fig. 4e, g and Supplementary Table 6).”

Discussion

Lines 322-324: “Because [...]” This is not a complete sentence.

We apologize for the grammatical error. We have now corrected this.

“But on the other hand, **since the distance between EBP and OBP is longer in DRD3 than in DRD2 (SEBP-OBP) and DRD4 (EBP-OBP)**, highly DRD3 selective compounds, such as R-22, can be obtained by designing compounds with a longer linker between the OBP binding moiety and EBP binding moiety.”

Figures

Figure 4: “optically pure enantiomers” is an unusual expression. The authors could use “individual enantiomers” or “enantiopure compounds” instead.

We thank the reviewer for pointing out our unsuitable expression. We have now replaced “optically pure enantiomers” with “**individual enantiomers**”.

“Figure 4. Comparison of functional activity of **individual enantiomers-O₄RE₆** and **O₄SE₆**.”

Figure 5: The title appears misleading to me, since the figure shows mostly assay data and no structure-based or mechanism-driven design.

We apologize for the misleading title. We have now corrected that by deleting the words “structure-based” and “mechanism-driven”, instead, we highlighted the functional activity validation.

“Figure 5. **Design of DRD2 selective agonist and validation of its functional activity.**”

Supplementary Figure S7: The figure shows docking of ORE and OSE compounds and not OLE compounds. Please correct the title.

We apologize for our incorrect writing. We have now corrected this.

“Supplementary Figure 7. Docking pose of O₄RE₆, O₄SE₆, O₇RE₆, O₇SE₆, O₈RE₆ and O₈SE₆.”

Reviewer #2 (Remarks to the Author):

The authors provided satisfactory explanations in part to my comments.

We thank the reviewer for the appreciation. We have made corrections according to the additional comments and we believe the revised manuscript is suitable for publication.

Several additional comments:

- I cannot find a ligplot analyses for haloperidol to see all potential hydrogen-binding and hydrophobic interactions in a -real- plot (SI3f is not satisfactory. There are no differences between van der Waals contacts or H-bonds).

As suggested by the reviewer, we changed the **Supplementary Fig. 3f** and highlighted the potential hydrogen bond.

- I suggest using a simulated-annealing omitted map at 3.0 sigma level, which is a default level for a fo-fc map type to control the quality of density.

- Have the authors even used a simulated-annealing omitted map? They call it fo-fc omit map which could be the wrong one. With the simulated-annealing omitted map the reader could see an almost unbiased map, which is essential for the contact analyses.

- The authors should make it clearer that there is no direct contact for the Cl of chlorobenzene moiety in haloperidol. This is clearly visible in the omitted map due to lack of density.

We thank the reviewer for this suggestion. As suggested by the reviewer, we have used the unbiased simulated-annealing omitted map at 3.0 sigma level and changed the **Supplementary Fig. 1c, d**. And, we also highlighted the density issue in our revised manuscript.

“Notably, the conserved residue Trp100^{EL1} in DRD2/haloperidol structure rotates outward away from the binding pocket as compared to the risperidone-bound structure (Fig. 1d-f). Similar crystal contacts between the extracellular tip of TM3 and the symmetry-related T4L at the DRD2/haloperidol and DRD2/risperidone crystal structures were observed, but there is no crystal contacts between Trp100^{EL1} and the symmetry-related T4L at both structures (Supplementary Fig. 2a, b). Therefore, the rotation of Trp100^{EL1} at DRD2 is unlikely induced by crystal packing forces. **Although the electron-density omit map was murky at the chlorobenzene moiety of the haloperidol (Supplementary Fig. 1c, d)**, haloperidol apparently prevents the inward rotation of Trp100^{EL1} (Fig. 1d, e), which may explain the difference between the two structures. And, the mutations of Trp100^{EL1} to Phe or Ala in DRD2 decreased the binding affinity of the haloperidol and L-741626 (Supplementary Table 2).”

- Regrading to the comment of Reviewer 1: I cannot see any crystal contacts in Supplementary Figure 2. "Crystal packing of D2-like receptors". Unfortunately, this figure does not give this statement. Is there any crystal contact (hydrophobic or H-bonds) in the whole ECL1 or at the end of TM2, TM1 or TM3 to the symmetry-related T4L or other parts of the protein construct? If so, the authors should clarify this in the main text. That would be fair to the reader. I also clearly see the differences in ECL1 between DRD2/haloperidol and DRD2/risperidone complexes, but a crystal contact can cause a specific local state (which does not necessarily mean that it has to be artificial). The same is true for the situation with DRD2/risperidone. Is there a crystal contact in the above region? In the manuscript is only a statement like “is unlikely induced by the crystal packing forces”.

We thank the reviewer for this suggestion. There are crystal contacts between extracellular tip of TM3 and the symmetry-related T4L at the DRD2/haloperidol and DRD2/risperidone crystal structures (Supplementary Fig. 2a, b). But, the crystal contacts are similar at both structures (Supplementary Fig. 2a, b). And, there is no crystal contacts between Trp100^{EL1} and the symmetry-related T4L at both structures (Supplementary Fig. 2a, b). Then, we believe the rotation of Trp100^{EL1} at DRD2 is unlikely

induced by crystal packing forces. For a better knowledge of the information, we have included the description of the crystal contacts.

“Notably, the conserved residue Trp100^{EL1} in DRD2/haloperidol structure rotates outward away from the binding pocket as compared to the risperidone-bound structure (Fig. 1d-f). **Similar crystal contacts between the extracellular tip of TM3 and the symmetry-related T4L at the DRD2/haloperidol and DRD2/risperidone crystal structures were observed, but there is no crystal contact between Trp100^{EL1} and the symmetry-related T4L at both structures (Supplementary Fig. 2a, b).** Therefore, the rotation of Trp100^{EL1} at DRD2 is unlikely induced by crystal packing forces. Although the electron-density omit map was murky at the chlorobenzene moiety of the haloperidol (Supplementary Fig. 1c, d), haloperidol apparently prevents the inward rotation of Trp100^{EL1} (Fig. 1d, e), which may explain the difference between the two structures. And, the mutations of Trp100^{EL1} to Phe or Ala in DRD2 decreased the binding affinity of the haloperidol and L-741626 (Supplementary Table 2).”

- I suggest to include the mutant data for the variants DRD2 F110W and F110Y in the manuscript. These data underline the fact that the size of the binding pocket, which is reduced by amino acids larger than F, is crucial, reflected in a decreased K_i for L-741.626 and haloperidol. It becomes really interesting when you compare DRD2 F1103.28C and F1103.28E. Both amino acids have rather similar sizes but completely different properties. Both mutants have almost similar K_i as the WT for both ligands. Here only the size of the amino acid seems to play a role for the binding affinity. This statement is finally strengthened with the variant DRD2 F1103.28A (increased K_i compared to the WT). These data simply belong at least in the supplement.

We thank the reviewer for the suggestion. Considering the reviewer's suggestion, we have re-phrased this paragraph and included the mutant data for the variants DRD2 F110^{3.28}W, F110^{3.28}Y, F110^{3.28}C and F110^{3.28}E in our main text and in the supplement (**Supplementary Table 2**).

“To further identify the key residue(s) responsible for the binding of DRD2-preferring compounds, we performed mutagenesis studies on the SEBP-related residues (Supplementary Table 2 and Supplementary Fig. 3f). The alanine substitution of most SEBP residues, except Phe110^{3.28}, slightly reduced the affinity of both haloperidol and L-741626 (Supplementary Table 2). In contrast, the mutation of Phe110^{3.28} to Ala or Leu on DRD2 greatly enhanced the binding of haloperidol or L-741626 (15.33 or 1.77-fold for haloperidol and 144.18 or 18.65-fold for L-741626) (Supplementary Table 2) , **while the mutation of Phe110^{3.28} to Trp or Tyr greatly reduced the binding of haloperidol and L-741626 (28.00 or 41.41-fold for haloperidol and 8.51 or 57.54-fold for L-741626) (Supplementary Table 2).** Furthermore, the mutation of Phe110^{3.28} to Cys or Glu, both of which have similar sizes with each other but with different physical properties, slightly influenced the binding affinity of both ligands, ruling out the possibility that the property of the amino acid affects ligand binding (Supplementary Table 2). It is possible that the alanine or leucine substitution of Phe110^{3.28} makes additional space for the DRD2 SEBP, facilitating the accommodation of the chlorobenzene moiety of haloperidol or L-741626 (Fig. 1b, e and 2a, e). And, the previous published studies already showed that the mutation of Phe110^{3.28} to Ala on DRD2 didn't enhance the binding of non-selective compounds--risperidone and nemonapride¹⁵. In summary, the size of the residue 3.28 seems to play a key role for the binding affinity of haloperidol and L-741626”

- The DRD2 W100ECL1F variant is also very interesting. It gives a hint that L-741.626 and haloperidol are in a slightly different conformation in the binding pocket (shallow vs deep?). The 28-fold decreased K_i in DRD2 W100ECL1F with haloperidol (compared to the WT) is an effect, which is partly seen with L-741.626 (8-fold). However, when the mutant DRD2 W100ECL1A is used, both ligands behave differently. With L-741.626 the K_i is 6-fold lowered similar to the DRD2 W100ECL1F variant, but with haloperidol only 3-fold. These data should also be mentioned.

We thank the reviewer for the suggestion. We have included the mutant data for the variants DRD2 W100^{EL1}A and W100^{EL1}F in our main text and in the supplement (**Supplementary Table 2**).

“Notably, the conserved residue Trp100^{EL1} in DRD2/haloperidol structure rotates outward away from the binding pocket as compared to the risperidone-bound structure (Fig. 1d-f). Similar crystal contacts

between the extracellular tip of TM3 and the symmetry-related T4L at the DRD2/haloperidol and DRD2/risperidone crystal structures were observed, but there is no crystal contact between Trp100^{EL1} and the symmetry-related T4L at both structures (Supplementary Fig. 2a, b). Therefore, the rotation of Trp100^{EL1} at DRD2 is unlikely induced by crystal packing forces. Although the electron-density omit map was murky at the chlorobenzene moiety of the haloperidol (Supplementary Fig. 1c, d), haloperidol apparently prevents the inward rotation of Trp100^{EL1} (Fig. 1d, e), which may explain the difference between the two structures. **And, the mutations of Trp100^{EL1} to Phe or Ala in DRD2 decreased the binding affinity of the haloperidol and L-741626 (Supplementary Table 2)."**

Reviewers' Comments:

Reviewer #2:

Remarks to the Author:

- After inspection of the crystal structure and its electron density (Many thanks for that!) I agree with the authors that the electron density is sufficient for proper fitting of the ligand haloperidol. Again, the resolution is only 3.1 Angstrom, i.e. no better electron density can be expected. The density is not really good and I recommend to describe it in one sentence (but please not as "murky"). One should also mention the "RSCC - real space correlation coefficient" value for the ligand (not the "RSRZ") in the methods part (>90% for this ligand is ok).
- I strongly recommend the authors to include all results and mutations of table A (reply letter) in the manuscript without exception and to discuss their consequences.
- All other of my remarks were sufficiently answered by the authors.

Response to Reviewers' Comments are in blue.

Reviewer #2 (Remarks to the Author):

- After inspection of the crystal structure and its electron density (Many thanks for that!) I agree with the authors that the electron density is sufficient for proper fitting of the ligand haloperidol. Again, the resolution is only 3.1 Angstrom, i.e. no better electron density can be expected. The density is not really good and I recommend to describe it in one sentence (but please not as "murky").

We thank the reviewer for pointing out our inadequate description. We have now changed it as follows:

“Although the electron-density omit map **partially missed** at the chlorobenzene moiety of the haloperidol (Supplementary Fig. 1c, d), haloperidol apparently prevents the inward rotation of Trp100^{EL1} (Fig. 1d, e), which may explain the difference between the two structures.”

One should also mention the "RSCC - real space correlation coefficient" value for the ligand (not the "RSRZ") in the methods part (>90% for this ligand is ok).

We thank the reviewer for pointing this out. We have now included a more detailed information of the ligand refinement at the methods.

“After the refinement, the RSCC (real space correlation coefficient) value of the haloperidol is 0.94, which means the electron density is proper fitting of the ligand haloperidol.”

- I strongly recommend the authors to include all results and mutations of table A (reply letter) in the manuscript without exception and to discuss their consequences.

We thank the reviewer for the suggestion. We have included the mutant data for the variants DRD2 in our main text and in the supplement (**Supplementary Table 2**).

“To further identify the key residue(s) responsible for the binding of DRD2-preferring compounds, we performed mutagenesis studies on the SEBP-related residues (Supplementary Table 2 and Supplementary Fig. 3f). The alanine substitution of most SEBP residues, except Phe110^{3,28}, slightly reduced the affinity of both haloperidol and L-741626 (Supplementary Table 2). In contrast, the mutation of Phe110^{3,28} to Ala or Leu on DRD2 greatly enhanced the binding of haloperidol or L-741626 (15.33 or 1.77-fold for haloperidol and 144.18 or 18.65-fold for L-741626) (Supplementary Table 2), **while the mutation of Phe110^{3,28} to Trp or Tyr greatly reduced the binding of haloperidol and L-741626 (28.00 or 41.41-fold for haloperidol and 8.51 or 57.54-fold for L-741626) (Supplementary Table 2). Furthermore, the mutation of Phe110^{3,28} to Cys or Glu, both of which have similar sizes with each other but with different physical properties, slightly influenced the binding affinity of both ligands, ruling out the possibility that the property of the amino acid affects ligand binding (Supplementary Table 2).** It is possible that the alanine or leucine substitution of Phe110^{3,28} makes additional space for the DRD2 SEBP, facilitating the accommodation of the chlorobenzene moiety of haloperidol or L-741626 (Fig. 1b, e and 2a, e). And, the previous published studies already showed that the mutation of Phe110^{3,28} to Ala on DRD2 didn't enhance the binding of non-selective compounds--risperidone and nemonapride¹⁵. In summary, the size of the residue 3.28 seems to play a key role for the binding affinity of haloperidol and L-741626”

“Notably, the conserved residue Trp100^{EL1} in DRD2/haloperidol structure rotates outward away from the binding pocket as compared to the risperidone-bound structure (Fig. 1d-f). Similar crystal contacts

between the extracellular tip of TM3 and the symmetry-related T4L at the DRD2/haloperidol and DRD2/risperidone crystal structures were observed, but there is no crystal contact between Trp100^{EL1} and the symmetry-related T4L at both structures (Supplementary Fig. 2a, b). Therefore, the rotation of Trp100^{EL1} at DRD2 is unlikely induced by crystal packing forces. Although the electron-density omit map was murky at the chlorobenzene moiety of the haloperidol (Supplementary Fig. 1c, d), haloperidol apparently prevents the inward rotation of Trp100^{EL1} (Fig. 1d, e), which may explain the difference between the two structures. **And, the mutations of Trp100^{EL1} to Phe or Ala in DRD2 decreased the binding affinity of the haloperidol and L-741626 (Supplementary Table 2)."**

- All other of my remarks were sufficiently answered by the authors.

We thank the reviewer for these positive comments. We have made corrections according to the additional comments and we believe the revised manuscript is suitable for publication.